# MATRIX MANIFOLD NEURAL NETWORKS++

**Xuan Son Nguyen, Shuo Yang, Aymeric Histace**
ETIS, UMR 8051, CY Cergy Paris University, ENSEA, CNRS, France
{xuan-son.nguyen,shuo.yang,aymeric.histace}@ensea.fr

## ABSTRACT

Deep neural networks (DNNs) on Riemannian manifolds have garnered increasing interest in various applied areas. For instance, DNNs on spherical and hyperbolic manifolds have been designed to solve a wide range of computer vision and nature language processing tasks. One of the key factors that contribute to the success of these networks is that spherical and hyperbolic manifolds have the rich algebraic structures of gyrogroups and gyrovector spaces. This enables principled and effective generalizations of the most successful DNNs to these manifolds. Recently, some works have shown that many concepts in the theory of gyrogroups and gyrovector spaces can also be generalized to matrix manifolds such as Symmetric Positive Definite (SPD) and Grassmann manifolds. As a result, some building blocks for SPD and Grassmann neural networks, e.g., isometric models and multinomial logistic regression (MLR) can be derived in a way that is fully analogous to their spherical and hyperbolic counterparts. Building upon these works, we design fully-connected (FC) and convolutional layers for SPD neural networks. We also develop MLR on Symmetric Positive Semi-definite (SPSD) manifolds, and propose a method for performing backpropagation with the Grassmann logarithmic map in the projector perspective. We demonstrate the effectiveness of the proposed approach in the human action recognition and node classification tasks.

## 1 INTRODUCTION

In recent years, deep neural networks on Riemannian manifolds have achieved impressive performance in many applications (Ganea et al., 2018; Skopek et al., 2020; Cruceru et al., 2021; Shimizu et al., 2021). The most popular neural networks in this family operate on hyperbolic spaces. Such spaces of constant sectional curvature, like spherical spaces, have the rich algebraic structure of gyrovector spaces. The theory of gyrovector spaces (Ungar, 2002; 2005; 2014) offers an elegant and powerful framework based on which natural generalizations (Ganea et al., 2018; Shimizu et al., 2021) of essential building blocks in DNNs are constructed for hyperbolic neural networks (HNNs).

Matrix manifolds such as SPD and Grassmann manifolds offer a convenient trade-off between structural richness and computational tractability (Cruceru et al., 2021; López et al., 2021). Therefore, in many applications, neural networks on matrix manifolds are attractive alternatives to their hyperbolic counterparts. However, unlike the approaches in Ganea et al. (2018); Shimizu et al. (2021), most existing approaches for building SPD and Grassmann neural networks (Dong et al., 2017; Huang & Gool, 2017; Huang et al., 2018; Nguyen et al., 2019; Brooks et al., 2019; Nguyen, 2021; Wang et al., 2021) do not provide necessary techniques and mathematical tools to generalize a broad class of DNNs to the considered manifolds.

Recently, the authors of Kim (2020); Nguyen (2022b) have shown that SPD and Grassmann manifolds have the structure of gyrovector spaces or that of nonreductive gyrovector spaces (Nguyen, 2022b) that share remarkable analogies with gyrovector spaces. The work in Nguyen & Yang (2023) takes one step forward in that direction by generalizing several notions in gyrovector spaces, e.g., the inner product and gyrodistance (Ungar, 2014) to SPD and Grassmann manifolds. This allows one to characterize certain gyroisometries of these manifolds and to construct MLR on SPD manifolds.

Although some useful notions in gyrovector spaces have been generalized to SPD and Grassmann manifolds (Nguyen, 2022a;b; Nguyen & Yang, 2023) that set the stage for an effective way of building neural networks on these manifolds, many questions remain open. In this paper, we aim at

addressing some limitations of existing works using a gyrovector space approach. Our contributions can be summarized as follows:

1. We generalize FC and convolutional layers to the SPD manifold setting.

2. We propose a method for performing backpropagation with the Grassmann logarithmic map in the projector perspective (Bendokat et al., 2020) without resorting to any approximation schemes. We then show how to construct graph convolutional networks (GCNs) on Grassmann manifolds.

3. We develop MLR on SPSD manifolds.

4. We showcase our approach in the human action recognition and node classification tasks.

## 2 PRELIMINARIES

### 2.1 SPD MANIFOLDS

The space of $n \times n$ SPD matrices, when provided with some geometric structures like a Riemannian metric, forms SPD manifold $\mathrm{Sym}_n^+$ (Arsigny et al., 2005). Data lying on SPD manifolds are commonly encountered in various domains (Huang & Gool, 2017; Brooks et al., 2019; Nguyen, 2021; Sukthanker et al., 2021; Nguyen, 2022b;a; Nguyen & Yang, 2023). In many applications, the use of Euclidean calculus on SPD manifolds often leads to unsatisfactory results (Arsigny et al., 2005). To tackle this issue, many Riemannian structures for SPD manifolds have been introduced. In this work, we focus on two widely used Riemannian metrics, i.e., Affine-Invariant (AI) (Pennec et al., 2004) and Log-Euclidean (LE) (Arsigny et al., 2005) metrics, and a recently introduced Riemannian metrics, i.e. Log-Cholesky (LC) metrics (Lin, 2019) that offer some advantages over Affine-Invariant and Log-Euclidean metrics.

### 2.2 GRASSMANN MANIFOLDS

Grassmann manifolds $\mathrm{Gr}_{n,p}$ are the collection of linear subspaces of fixed dimension $p$ of the Euclidean space $\mathbb{R}^n$ (Edelman et al., 1998). Data lying on Grassmann manifolds arise naturally in many applications (Absil et al., 2007; Bendokat et al., 2020). Points on Grassmann manifolds can be represented from different perspectives (Bendokat et al., 2020). Two typical approaches use projection matrices or those with orthonormal columns. Each of them can be effective in some problems but might be inappropriate in some other contexts (Nguyen, 2022b). Although geometrical descriptions of Grassmann manifolds have been given in numerous works (Edelman et al., 1998), some computational issues remain to be addressed. For instance, the question of how to effectively perform backpropagation with the Grassmann logarithmic map in the projector perspective remains open.

### 2.3 NEURAL NETWORKS ON SPD AND GRASSMANN MANIFOLDS

#### 2.3.1 NEURAL NETWORKS ON SPD MANIFOLDS

The work in Huang & Gool (2017) introduces SPDNet with three novel layers, i.e., Bimap, LogEig, and ReEig layers that has become one of the most successful architectures in the field. In Brooks et al. (2019), the authors further improve SPDNet by developing Riemannian versions of batch normalization layers. Following these works, some works (Nguyen et al., 2019; Nguyen, 2021; Wang et al., 2021; Kobler et al., 2022; Ju & Guan, 2023) design variants of Bimap and batch normalization layers in SPD neural networks. The work in Chakraborty et al. (2020) presents a different approach based on intrinsic operations on SPD manifolds. Their proposed layers have nice theoretical properties. A common limitation of the above works is that they do not provide necessary mathematical tools for constructing many essential building blocks of DNNs on SPD manifolds. Recently, some works (Nguyen, 2022a;b; Nguyen & Yang, 2023) take a gyrovector space approach that enables natural generalizations of some building blocks of DNNs, e.g., MLR for SPD neural networks.

#### 2.3.2 NEURAL NETWORKS ON GRASSMANN MANIFOLDS

In Huang et al. (2018), the authors propose GrNet that explores the same rule of matrix backpropagation (Ionescu et al., 2015) as SPDNet. Some existing works (Wang & Wu, 2020; Souza et al.,

2020) are also inspired by GrNet. Like their SPD counterparts, most existing Grassmann neural networks are not built upon a mathematical framework that allows one to generalize a broad class of DNNs to Grassmann manifolds. Using a gyrovector space approach, Nguyen & Yang (2023) has shown that some concepts in Euclidean spaces can be naturally extended to Grassmann manifolds.

## 3 PROPOSED APPROACH

### 3.1 NOTATION

Let $\mathcal{M}$ be a homogeneous Riemannian manifold, $T_{\mathbf{P}}\mathcal{M}$ be the tangent space of $\mathcal{M}$ at $\mathbf{P} \in \mathcal{M}$. Denote by $\exp(\mathbf{P})$ and $\log(\mathbf{P})$ the usual matrix exponential and logarithm of $\mathbf{P}$, $\mathrm{Exp}_{\mathbf{P}}(\mathbf{W})$ the exponential map at $\mathbf{P}$ that associates to a tangent vector $\mathbf{W} \in T_{\mathbf{P}}\mathcal{M}$ a point of $\mathcal{M}$, $\mathrm{Log}_{\mathbf{P}}(\mathbf{Q})$ the logarithmic map of $\mathbf{Q} \in \mathcal{M}$ at $\mathbf{P}$, $\mathcal{T}_{\mathbf{P}\to\mathbf{Q}}(\mathbf{W})$ the parallel transport of $\mathbf{W}$ from $\mathbf{P}$ to $\mathbf{Q}$ along geodesics connecting $\mathbf{P}$ and $\mathbf{Q}$. For simplicity of exposition, we will concentrate on real matrices. Denote by $\mathrm{M}_{n,m}$ the space of $n \times m$ matrices, $\mathrm{Sym}_n^+$ the space of $n \times n$ SPD matrices, $\mathrm{Sym}_n$ the space of $n \times n$ symmetric matrices, $\mathrm{S}_{n,p}^+$ the space of $n \times n$ SPSD matrices of rank $p \leq n$, $\mathrm{Gr}_{n,p}$ the $p$-dimensional subspaces of $\mathbb{R}^n$ in the projector perspective. For clarity of presentation, let $\widetilde{\mathrm{Gr}}_{n,p}$ be the $p$-dimensional subspaces of $\mathbb{R}^n$ in the ONB (orthonormal basis) perspective (Bendokat et al., 2020). For notations related to SPD manifolds, we use the letter $g \in \{ai, le, lc\}$ as a subscript (superscript) to indicate the considered Riemannian metric, unless otherwise stated. Other notations will be introduced in appropriate paragraphs. Our notations are summarized in Appendix A.

### 3.2 NEURAL NETWORKS ON SPD MANIFOLDS

In Nguyen (2022a;b), the author has shown that SPD manifolds with Affine-Invariant, Log-Euclidean, and Log-Cholesky metrics form gyrovector spaces referred to as AI, LE, and LC gyrovector spaces, respectively. We adopt the notations in these works and consider the case where $r = 1$ (see Nguyen (2022b), Definition 3.1). Let $\oplus_{ai}, \oplus_{le}$, and $\oplus_{lc}$ be the binary operations in AI, LE, and LC gyrovector spaces, respectively. Let $\ominus_{ai}, \ominus_{le}$, and $\ominus_{lc}$ be the inverse operations in AI, LE, and LC gyrovector spaces, respectively. These operations are given in Appendix G.

#### 3.2.1 FC LAYERS IN SPD NEURAL NETWORKS

Our method for generalizing FC layers to the SPD manifold setting relies on a reformulation of SPD hypergyroplanes (Nguyen & Yang, 2023). We first recap the definition of SPD hypergyroplanes.

**Definition 3.1** (**SPD Hypergyroplanes (Nguyen & Yang, 2023)**). *For* $\mathbf{P} \in \mathrm{Sym}_n^{+,g}$, $\mathbf{W} \in T_{\mathbf{P}}\mathrm{Sym}_n^{+,g}$, *SPD hypergyroplanes are defined as*

$$\mathcal{H}_{\mathbf{W},\mathbf{P}}^{spd,g} = \{\mathbf{Q} \in \mathrm{Sym}_n^{+,g} : \langle \mathrm{Log}_{\mathbf{P}}^g(\mathbf{Q}), \mathbf{W} \rangle_{\mathbf{P}}^g = 0\},$$

*where* $\langle .,. \rangle_{\mathbf{P}}^g$ *denotes the inner product at* $\mathbf{P}$ *given by the considered Riemannian metric.*

Proposition 3.2 gives an equivalent definition for SPD hypergyroplanes.

**Proposition 3.2.** *Let* $\mathbf{P} \in \mathrm{Sym}_n^{+,g}$, $\mathbf{W} \in T_{\mathbf{P}}\mathrm{Sym}_n^{+,g}$, *and* $\mathcal{H}_{\mathbf{W},\mathbf{P}}^{spd,g}$ *be the SPD hypergyroplanes defined in Definition 3.1. Then*

$$\mathcal{H}_{\mathbf{W},\mathbf{P}}^{spd,g} = \{\mathbf{Q} \in \mathrm{Sym}_n^{+,g} : \langle \ominus_g \mathbf{P} \oplus_g \mathbf{Q}, \mathrm{Exp}_{\mathbf{I}_n}^g(\mathcal{T}_{\mathbf{P}\to\mathbf{I}_n}^g(\mathbf{W})) \rangle^g = 0\},$$

*where* $\mathbf{I}_n$ *denotes the* $n \times n$ *identity matrix, and* $\langle .,. \rangle^g$ *is the SPD inner product in* $\mathrm{Sym}_n^{+,g}$ *(Nguyen & Yang, 2023) (see Appendix G.7 for the definition of the SPD inner product).*

**Proof** See Appendix I.

In DNNs, an FC layer linearly transforms the input in such a way that the $k$-th dimension of the output corresponds to the signed distance from the output to the hyperplane that contains the origin and is orthonormal to the $k$-th axis of the output space. This interpretation has proven useful in generalizing FC layers to the hyperbolic setting (Shimizu et al., 2021).

Notice that the equation of SPD hypergyroplanes in Proposition 3.2 has the form $\langle \ominus_g \mathbf{P} \oplus_g \mathbf{Q}, \mathbf{W}' \rangle^g = 0$, where $\mathbf{W}' \in \mathrm{Sym}_n^{+,g}$. This equation can be seen as a generalization of the hyperplane equation $\langle w, x \rangle + b = \langle -p + x, w \rangle = 0$, where $w, x, p \in \mathbb{R}^n, b \in \mathbb{R}$, and $\langle p, w \rangle = -b$. Therefore, Proposition 3.2 suggests that any linear function of an SPD matrix $\mathbf{X} \in \mathrm{Sym}_n^{+,g}$ can be written as $\langle \ominus_g \mathbf{P} \oplus_g \mathbf{X}, \mathbf{W} \rangle^g$, where $\mathbf{P}, \mathbf{W} \in \mathrm{Sym}_n^{+,g}$. The above interpretation of FC layers now can be applied to our case for constructing FC layers in SPD neural networks. For convenience of presentation, in Definition 3.3, we will index the dimensions (axes) of the output space using two subscripts corresponding to the row and column indices in a matrix.

**Definition 3.3.** *Let $E_{(i,j)}^g, i \leq j, i, j = 1, \ldots, m$ be the $(i,j)$-th axis of the output space. An SPD hypergyroplane that contains the origin and is orthonormal to the $E_{(i,j)}^g$ axis can be defined as*

$$\mathcal{H}_{\mathrm{Log}_{\mathbf{I}_m}^g(E_{(i,j)}^g), \mathbf{I}_m}^{spd,g} = \{\mathbf{Q} \in \mathrm{Sym}_m^{+,g} : \langle \mathbf{Q}, E_{(i,j)}^g \rangle^g = 0\}.$$

It remains to specify an orthonormal basis for each family of the considered Riemannian metrics of SPD manifolds. Proposition 3.4 gives such an orthonormal basis for AI gyrovector spaces along with the expression for the output of FC layers with Affine-Invariant metrics.

**Proposition 3.4** (FC layers with Affine-Invariant Metrics). *Let $(\mathbf{e}_1, \ldots, \mathbf{e}_m), \|\mathbf{e}_i\| = 1, i = 1 \ldots, m$ be an orthonormal basis of $\mathbb{R}^m$. Let $\langle ., . \rangle_{\mathbf{P}}^{ai}$ be the Affine-Invariant metric computed at $\mathbf{P} \in \mathrm{Sym}_m^{+,ai}$ as*

$$\langle \mathbf{V}, \mathbf{W} \rangle_{\mathbf{P}}^{ai} = \mathrm{Tr}(\mathbf{V}\mathbf{P}^{-1}\mathbf{W}\mathbf{P}^{-1}) + \beta \, \mathrm{Tr}(\mathbf{V}\mathbf{P}^{-1}) \, \mathrm{Tr}(\mathbf{W}\mathbf{P}^{-1}),$$

*where $\beta > -\frac{1}{m}$. An orthonormal basis $E_{(i,j)}^{ai}, i \leq j, i, j = 1, \ldots, m$ of $\mathrm{Sym}_m^{+,ai}$ can be given as*

$$E_{(i,j)}^{ai} = \begin{cases} \exp\left( \mathbf{e}_i \mathbf{e}_j^T - \frac{1}{m}\left(1 - \frac{1}{\sqrt{1+m\beta}}\right)\mathbf{I}_m \right), & \text{if } i = j \\ \exp\left( \frac{\mathbf{e}_i \mathbf{e}_j^T + \mathbf{e}_j \mathbf{e}_i^T}{\sqrt{2}} \right), & \text{if } i < j \end{cases}$$

*Denote by $v_{(i,j)}(\mathbf{X}) = \langle \ominus_{ai}\mathbf{P}_{(i,j)} \oplus_{ai} \mathbf{X}, \mathbf{W}_{(i,j)} \rangle^{ai}, \mathbf{P}_{(i,j)}, \mathbf{W}_{(i,j)} \in \mathrm{Sym}_n^{+,ai}, i \leq j, i, j = 1, \ldots, m$. Let $\alpha = \frac{1}{m}(\sqrt{1+m\beta} - 1)$. Then the output of an FC layer is computed as $\mathbf{Y} = \exp\left([y_{(i,j)}]_{i,j=1}^m\right)$, where $[y_{(i,j)}]_{i,j=1}^m$ is the matrix having $y_{(i,j)}$ as the element at the $i$-th row and $j$-th column, and $y_{(i,j)}$ is given by*

$$y_{(i,j)} = \begin{cases} v_{(i,j)}(\mathbf{X}) + \alpha \sum_{k=1}^m v_{(k,k)}(\mathbf{X}), & \text{if } i = j \\ \frac{1}{\sqrt{2}}v_{(i,j)}(\mathbf{X}), & \text{if } i < j \\ \frac{1}{\sqrt{2}}v_{(j,i)}(\mathbf{X}), & \text{if } i > j \end{cases}$$

**Proof**   See Appendix J.

As shown in Arsigny et al. (2005), a Log-Euclidean metric on $\mathrm{Sym}_n^{+,le}$ can be obtained from any inner product on $\mathrm{Sym}_n$. In this work, we consider a metric that is invariant under all similarity transformations, i.e., the metric $\langle \mathbf{W}, \mathbf{V} \rangle_{\mathbf{I}_n}^{le} = \mathrm{Tr}(\mathbf{W}\mathbf{V})$. We have the following result.

**Proposition 3.5** (FC layers with Log-Euclidean Metrics). *An orthonormal basis $E_{(i,j)}^{le}, i \leq j, i, j = 1, \ldots, m$ of $\mathrm{Sym}_m^{+,le}$ can be given by*

$$E_{(i,j)}^{le} = \begin{cases} \exp\left( \mathbf{e}_i \mathbf{e}_j^T \right), & \text{if } i = j \\ \exp\left( \frac{\mathbf{e}_i \mathbf{e}_j^T + \mathbf{e}_j \mathbf{e}_i^T}{\sqrt{2}} \right), & \text{if } i < j \end{cases}$$

*Let $v_{(i,j)}(\mathbf{X}) = \langle \ominus_{le}\mathbf{P}_{(i,j)} \oplus_{le} \mathbf{X}, \mathbf{W}_{(i,j)} \rangle^{le}, \mathbf{P}_{(i,j)}, \mathbf{W}_{(i,j)} \in \mathrm{Sym}_n^{+,le}, i \leq j, i, j = 1, \ldots, m$. Then the output of an FC layer is computed as $\mathbf{Y} = \exp\left([y_{(i,j)}]_{i,j=1}^m\right)$, where $y_{(i,j)}$ is given by*

$$y_{(i,j)} = \begin{cases} v_{(i,j)}(\mathbf{X}), & \text{if } i = j \\ \frac{1}{\sqrt{2}}v_{(i,j)}(\mathbf{X}), & \text{if } i < j \\ \frac{1}{\sqrt{2}}v_{(j,i)}(\mathbf{X}), & \text{if } i > j \end{cases}$$

**Proof** See Appendix K.

Finally, we give the characterization of an orthonormal basis for LC gyrovector spaces and the expression for the output of FC layers with Log-Cholesky metrics.

**Proposition 3.6** (**FC layers with Log-Cholesky Metrics**). *An orthonormal basis* $E_{(i,j)}^{lc}, i \leq j, i, j = 1, \ldots, m$ *of* $\mathrm{Sym}_m^{+,lc}$ *can be given by*

$$
E_{(i,j)}^{lc} = \begin{cases} (e-1)\mathbf{e}_i\mathbf{e}_j^T + \mathbf{I}_m, & \text{if } i = j \\ (\mathbf{e}_j\mathbf{e}_i^T + \mathbf{I}_m)(\mathbf{e}_i\mathbf{e}_j^T + \mathbf{I}_m), & \text{if } i < j \end{cases}
$$

*Let* $v_{(i,j)}(\mathbf{X}) = \langle \ominus_{lc}\mathbf{P}_{(i,j)} \oplus_{lc} \mathbf{X}, \mathbf{W}_{(i,j)} \rangle^{lc}, \mathbf{P}_{(i,j)}, \mathbf{W}_{(i,j)} \in \mathrm{Sym}_n^{+,lc}, i \leq j, i, j = 1, \ldots, m.$ *Then the output of an FC layer is computed as* $\mathbf{Y} = \overline{\mathbf{Y}}\overline{\mathbf{Y}}^T$, *where* $\overline{\mathbf{Y}} = [y_{(i,j)}]_{i,j=1}^m$, *and* $y_{(i,j)}$ *is given by*

$$
y_{(j,i)} = \begin{cases} \exp(v_{(i,j)}(\mathbf{X})), & \text{if } i = j \\ v_{(i,j)}(\mathbf{X}), & \text{if } i < j \\ 0, & \text{if } i > j \end{cases}
$$

**Proof** See Appendix L.

### 3.2.2 CONVOLUTIONAL LAYERS IN SPD NEURAL NETWORKS

Consider applying a 2D convolutional layer to a multi-channel image. Let $N_{in}$ and $N_{out}$ be the numbers of input and output channels, respectively. Denote by $y_{(i,j)}^k, i = 1, \ldots, N_{row}, j = 1, \ldots, N_{col}, k = 1, \ldots, N_{out}$ the value of the $k$-th output channel at pixel $(i,j)$. Then

$$
y_{(i,j)}^k = \sum_{l=1}^{N_{in}} \langle \mathbf{w}^{(l,k)}, \mathbf{x}_{(i,j)}^l \rangle + b^k, \tag{1}
$$

where $\mathbf{x}_{(i,j)}^l$ is a receptive field of the $l$-th input channel, $\mathbf{w}^{(l,k)}$ is the filter associated with the $l$-th input channel and the $k$-th output channel, and $b^k$ is the bias for the $k$-th output channel. Let $\mathbf{X}_{(i,j)} = \mathrm{concat}(\mathbf{x}_{(i,j)}^1, \ldots, \mathbf{x}_{(i,j)}^{N_{in}})$, $\mathbf{W}_k = \mathrm{concat}(\mathbf{w}^{(1,k)}, \ldots, \mathbf{w}^{(N_{in},k)})$, where operation $\mathrm{concat}(.)$ concatenates all of its arguments. Then Eq. (1) can be rewritten (Shimizu et al., 2021) as

$$
y_{(i,j)}^k = \langle \mathbf{W}_k, \mathbf{X}_{(i,j)} \rangle + b^k. \tag{2}
$$

Note that Eq. (2) has the form $\langle w, x \rangle + b$ and thus the computations discussed in Section 3.2.1 can be applied to implement convolutional layers in SPD neural networks. Specifically, given a set of SPD matrices $\mathbf{P}_i \in \mathrm{Sym}_n^{+,g}, i = 1, \ldots, N$, operation $\mathrm{concat}_{spd}(\mathbf{P}_1, \ldots, \mathbf{P}_N)$ produces a block diagonal matrix having $\mathbf{P}_i$ as diagonal elements.

In Chakraborty et al. (2020), the authors design a convolution operation for SPD neural networks. However, their method is based on the concept of weighted Fréchet Mean, while ours is built upon the concepts of SPD hypergyroplane and SPD pseudo-gyrodistance from an SPD matrix to an SPD hypergyroplane (Nguyen & Yang, 2023). Also, our convolution operation can be used for dimensionality reduction, while theirs always produces an output of the same dimension as the inputs.

### 3.3 MLR IN STRUCTURE SPACES

Motivated by the works in Nguyen (2022a); Nguyen & Yang (2023), in this section, we aim to build MLR on SPSD manifolds. For any $\mathbf{P} \in \mathrm{S}_{n,p}^+$, we consider the decomposition $\mathbf{P} = \mathbf{U}_P\mathbf{S}_P\mathbf{U}_P^T$, where $\mathbf{U}_P \in \widetilde{\mathrm{Gr}}_{n,p}$ and $\mathbf{S}_P \in \mathrm{Sym}_p^+$. Each element of $\mathrm{S}_{n,p}^+$ can be seen as a flat $p$-dimensional ellipsoid in $\mathbb{R}^n$ (Bonnabel et al., 2013). The flat ellipsoid belongs to a $p$-dimensional subspace spanned by the columns of $\mathbf{U}_P$, while the $p \times p$ SPD matrix $\mathbf{S}_P$ defines the shape of the ellipsoid in $\mathrm{Sym}_p^+$. A canonical representation of $\mathbf{P}$ in structure space $\widetilde{\mathrm{Gr}}_{n,p} \times \mathrm{Sym}_p^+$ is computed by identifying a common subspace and then rotating $\mathbf{U}_P$ to this subspace. The SPD matrix $\mathbf{S}_P$ is rotated accordingly to reflect the changes of $\mathbf{U}_P$. Details of these computations are given in Appendix H.

Assuming that a canonical representation in structure space $\widetilde{\mathrm{Gr}}_{n,p} \times \mathrm{Sym}_p^+$ is obtained for each point in $\mathrm{S}_{n,p}^+$, we now discuss how to build MLR in this space. As one of the first steps for developing network building blocks in a gyrovector space approach is to construct some basic operations in the considered manifold, we give the definitions of the binary and inverse operations in the following.

**Definition 3.7 (The Binary Operation in Structure Spaces).** *Let* $(\mathbf{U}_P, \mathbf{S}_P), (\mathbf{U}_Q, \mathbf{S}_Q) \in \widetilde{\mathrm{Gr}}_{n,p} \times \mathrm{Sym}_p^{+,g}$. *Then the binary operation* $\oplus_{psd,g}$ *in structure space* $\widetilde{\mathrm{Gr}}_{n,p} \times \mathrm{Sym}_p^{+,g}$ *is defined as*

$$(\mathbf{U}_P, \mathbf{S}_P) \oplus_{psd,g} (\mathbf{U}_Q, \mathbf{S}_Q) = (\mathbf{U}_P \widetilde{\oplus}_{gr} \mathbf{U}_Q, \mathbf{S}_P \oplus_g \mathbf{S}_Q),$$

*where* $\widetilde{\oplus}_{gr}$ *is the binary operation in* $\widetilde{\mathrm{Gr}}_{n,p}$ *(see Appendix G.6 for the definition of* $\widetilde{\oplus}_{gr}$*).*

**Definition 3.8 (The Inverse Operation in Structure Spaces).** *Let* $(\mathbf{U}_P, \mathbf{S}_P) \in \widetilde{\mathrm{Gr}}_{n,p} \times \mathrm{Sym}_p^{+,g}$. *Then the inverse operation* $\ominus_{psd,g}$ *in structure space* $\widetilde{\mathrm{Gr}}_{n,p} \times \mathrm{Sym}_p^{+,g}$ *is defined as*

$$\ominus_{psd,g}(\mathbf{U}_P, \mathbf{S}_P) = (\widetilde{\ominus}_{gr}\mathbf{U}_P, \ominus_g\mathbf{S}_P),$$

*where* $\widetilde{\ominus}_{gr}$ *is the inverse operation in* $\widetilde{\mathrm{Gr}}_{n,p}$ *(see Appendix G.6 for the definition of* $\widetilde{\ominus}_{gr}$*).*

Our construction of the binary and inverse operations in $\widetilde{\mathrm{Gr}}_{n,p} \times \mathrm{Sym}_p^{+,g}$ is clearly advantageous compared to the method in Nguyen (2022a) since this method does not preserve the information about the subspaces of the terms involved in these operations. In addition to the binary and inverse operations, we also need to define the inner product in structure spaces.

**Definition 3.9 (The Inner Product in Structure Spaces).** *Let* $(\mathbf{U}_P, \mathbf{S}_P), (\mathbf{U}_Q, \mathbf{S}_Q) \in \widetilde{\mathrm{Gr}}_{n,p} \times \mathrm{Sym}_p^{+,g}$. *Then the inner product in structure space* $\widetilde{\mathrm{Gr}}_{n,p} \times \mathrm{Sym}_p^{+,g}$ *is defined as*

$$\langle(\mathbf{U}_P, \mathbf{S}_P), (\mathbf{U}_Q, \mathbf{S}_Q)\rangle^{psd,g} = \lambda\langle\mathbf{U}_P\mathbf{U}_P^T, \mathbf{U}_Q\mathbf{U}_Q^T\rangle^{gr} + \langle\mathbf{S}_P, \mathbf{S}_Q\rangle^g,$$

*where* $\lambda > 0$, $\langle., .\rangle^{gr}$ *is the Grassmann inner product (Nguyen & Yang, 2023) (see Appendix G.7).*

The key idea to generalize MLR to a Riemannian manifold is to change the margin to reflect the geometry of the considered manifold (a formulation of MLR from the perspective of distances to hyperplanes is given in Appendix C). This requires the notions of hyperplanes and margin in the considered manifold that are referred to as hypergyroplanes and pseudo-gyrodistances (Nguyen & Yang, 2023), respectively. In our case, the definition of hypergyroplanes in structure spaces, suggested by Proposition 3.2, can be given below.

**Definition 3.10 (Hypergyroplanes in Structure Spaces).** *Let* $\mathbf{P}, \mathbf{W} \in \widetilde{\mathrm{Gr}}_{n,p} \times \mathrm{Sym}_p^{+,g}$. *Then hypergyroplanes in structure space* $\widetilde{\mathrm{Gr}}_{n,p} \times \mathrm{Sym}_p^{+,g}$ *are defined as*

$$\mathcal{H}_{\mathbf{W}, \mathbf{P}}^{psd,g} = \{\mathbf{Q} \in \widetilde{\mathrm{Gr}}_{n,p} \times \mathrm{Sym}_p^{+,g} : \langle\ominus_{psd,g}\mathbf{P} \oplus_{psd,g} \mathbf{Q}, \mathbf{W}\rangle^{psd,g} = 0\}.$$

Pseudo-gyrodistances in structure spaces can be defined in the same way as SPD pseudo-gyrodistances. We refer the reader to Appendix G.8 for all related notions. Theorem 3.11 gives an expression for the pseudo-gyrodistance from a point to a hypergyroplane in a structure space.

**Theorem 3.11 (Pseudo-gyrodistances in Structure Spaces).** *Let* $\mathbf{W} = (\mathbf{U}_W, \mathbf{S}_W)$, $\mathbf{P} = (\mathbf{U}_P, \mathbf{S}_P)$, $\mathbf{X} = (\mathbf{U}_X, \mathbf{S}_X) \in \widetilde{\mathrm{Gr}}_{n,p} \times \mathrm{Sym}_p^{+,g}$, *and* $\mathcal{H}_{\mathbf{W}, \mathbf{P}}^{psd,g}$ *be a hypergyroplane in structure space* $\widetilde{\mathrm{Gr}}_{n,p} \times \mathrm{Sym}_p^{+,g}$. *Then the pseudo-gyrodistance from* $\mathbf{X}$ *to* $\mathcal{H}_{\mathbf{W}, \mathbf{P}}^{psd,g}$ *is given by*

$$\bar{d}(\mathbf{X}, \mathcal{H}_{\mathbf{W}, \mathbf{P}}^{psd,g}) = \frac{|\lambda\langle(\widetilde{\ominus}_{gr}\mathbf{U}_P\widetilde{\oplus}_{gr}\mathbf{U}_X)(\widetilde{\ominus}_{gr}\mathbf{U}_P\widetilde{\oplus}_{gr}\mathbf{U}_X)^T, \mathbf{U}_W\mathbf{U}_W^T\rangle^{gr} + \langle\ominus_g\mathbf{S}_P \oplus_g \mathbf{S}_X, \mathbf{S}_W\rangle^g|}{\sqrt{\lambda(\|\mathbf{U}_W\mathbf{U}_W^T\|^{gr})^2 + (\|\mathbf{S}_W\|^g)^2}},$$

*where* $\|.\|^{gr}$ *and* $\|.\|^g$ *are the norms induced by the Grassmann and SPD inner products, respectively.*

**Proof** See Appendix M.

The algorithm for computing the pseudo-gyrodistances is given in Appendix B.

### 3.4 Neural Networks on Grassmann Manifolds

In this section, we present a method for computing the Grassmann logarithmic map in the projector perspective. We then propose GCNs on Grassmann manifolds.

#### 3.4.1 Grassmann Logarithmic Map in The Projector Perspective

The Grassmann logarithmic map is given (Batzies et al., 2015; Bendokat et al., 2020) by

$$\mathrm{Log}_{\mathbf{P}}^{gr}(\mathbf{Q}) = [\Omega, \mathbf{P}],$$

where $\mathbf{P}, \mathbf{Q} \in \mathrm{Gr}_{n,p}$, and $\Omega$ is computed as

$$\Omega = \frac{1}{2} \log \left( (\mathbf{I}_n - 2\mathbf{Q})(\mathbf{I}_n - 2\mathbf{P}) \right).$$

Notice that the matrix $(\mathbf{I}_n - 2\mathbf{Q})(\mathbf{I}_n - 2\mathbf{P})$ is generally not an SPD matrix. This raises an issue when one needs to implement an operation that requires the Grassmann logarithmic map in the projector perspective using popular deep learning frameworks like PyTorch and Tensorflow, since the matrix logarithm function is not differentiable in these frameworks. To deal with this issue, we rely on the following result that allows us to compute the Grassmann logarithmic map in the projector perspective from the Grassmann logarithmic map in the ONB perspective.

**Proposition 3.12.** *Let $\tau$ be the mapping such that*

$$\tau : \widetilde{\mathrm{Gr}}_{n,p} \to \mathrm{Gr}_{n,p}, \ \mathbf{U} \mapsto \mathbf{U}\mathbf{U}^T.$$

*Let $\widetilde{\mathrm{Log}}_{\mathbf{U}}^{gr}(\mathbf{V}), \mathbf{U}, \mathbf{V} \in \widetilde{\mathrm{Gr}}_{n,p}$ be the logarithmic map of $\mathbf{V}$ at $\mathbf{U}$ in the ONB perspective. Then*

$$\mathrm{Log}_{\mathbf{P}}^{gr}(\mathbf{Q}) = \tau^{-1}(\mathbf{P}) \left( \widetilde{\mathrm{Log}}_{\tau^{-1}(\mathbf{P})}^{gr}(\tau^{-1}(\mathbf{Q})) \right)^T + \widetilde{\mathrm{Log}}_{\tau^{-1}(\mathbf{P})}^{gr}(\tau^{-1}(\mathbf{Q}))\tau^{-1}(\mathbf{P})^T.$$

**Proof**   See Appendix N.

Note that the Grassmann logarithmic map $\widetilde{\mathrm{Log}}_{\mathbf{U}}^{gr}(\mathbf{V})$ can be computed via singular value decomposition (SVD) that is a differentiable operation in PyTorch and Tensorflow (see Appendix E.2.2). Therefore, Proposition 3.12 provides an effective implementation of the Grassmann logarithmic map in the projector perspective for gradient-based learning.

#### 3.4.2 Graph Convolutional Networks on Grassmann Manifolds

We propose to extend GCNs to Grassmann geometry using an approach similar to Chami et al. (2019); Zhao et al. (2023). Let $\mathcal{G} = (\mathcal{V}, \mathcal{E})$ be a graph with vertex set $\mathcal{V}$ and edge set $\mathcal{E}$, $\mathbf{x}_i^l, i \in \mathcal{V}$ be the embedding of node $i$ at layer $l$ ($l = 0$ indicates input node features), $\mathcal{N}(i) = \{j : (i,j) \in \mathcal{E}\}$ be the set of neighbors of $i \in \mathcal{V}$, $\mathbf{W}^l$ and $\mathbf{b}^l$ be the weight and bias for layer $l$, and $\sigma(.)$ be a non-linear activation function. A basic GCN message-passing update (Zhao et al., 2023) can be expressed as

$$\mathbf{p}_i^l = \mathbf{W}^l \mathbf{x}_i^{l-1} \qquad \text{(feature transformation)}$$

$$\mathbf{q}_i^l = \sum_{j \in \mathcal{N}(i)} w_{ij} \mathbf{p}_j^l \qquad \text{(aggregation)}$$

$$\mathbf{x}_i^l = \sigma(\mathbf{q}_i^l + \mathbf{b}^l) \qquad \text{(bias and nonlinearity)}$$

For the aggregation operation, the weights $w_{ij}$ can be computed using different methods (Kipf & Welling, 2017; Hamilton et al., 2017). Let $\mathbf{X}_i^l \in \mathrm{Gr}_{n,p}, i \in \mathcal{V}$ be the Grassmann embedding of node $i$ at layer $l$. For feature transformation on Grassmann manifolds, we use isometry maps based on left Grassmann gyrotranslations (Nguyen & Yang, 2023), i.e.,

$$\phi_{\mathbf{M}}(\mathbf{X}_i^l) = \exp([\mathrm{Log}_{\mathbf{I}_{n,p}}^{gr}(\mathbf{M}), \mathbf{I}_{n,p}])\mathbf{X}_i^l \exp(-[\mathrm{Log}_{\mathbf{I}_{n,p}}^{gr}(\mathbf{M}), \mathbf{I}_{n,p}]),$$

where $\mathbf{I}_{n,p} = \begin{bmatrix} \mathbf{I}_p & 0 \\ 0 & 0 \end{bmatrix} \in \mathrm{M}_{n,n}$, and $\mathbf{M} \in \mathrm{Gr}_{n,p}$ is a model parameter. Let $\mathrm{Exp}^{gr}(.)$ be the exponential map in $\mathrm{Gr}_{n,p}$. Then the aggregation process is performed as

$$\mathbf{Q}_i^l = \mathrm{Exp}_{\mathbf{I}_{n,p}}^{gr} \left( \sum_{j \in \mathcal{N}(i)} k_{i,j} \mathrm{Log}_{\mathbf{I}_{n,p}}^{gr}(\mathbf{P}_j^l) \right),$$

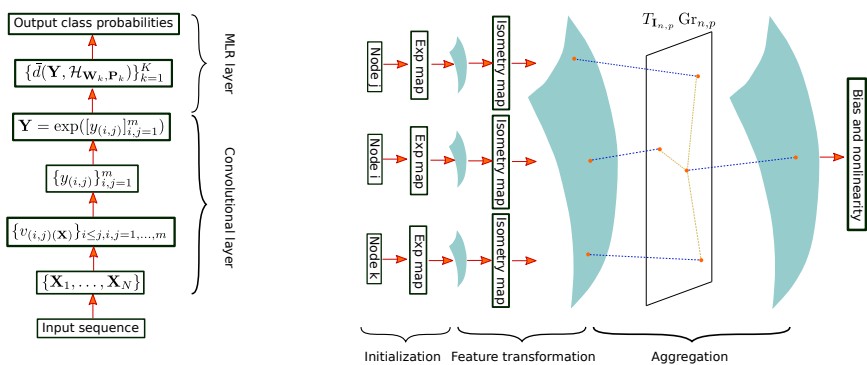

Figure 1: The pipelines of GyroSpd++ (left) and Gr-GCN++ (right).

where $\mathbf{P}_i^l$ and $\mathbf{Q}_i^l$ are the input and output node features of the aggregation operation, and $k_{i,j} = |\mathcal{N}(i)|^{-\frac{1}{2}}|\mathcal{N}(j)|^{-\frac{1}{2}}$ represents the relative importance of node $j$ to node $i$. For any $\mathbf{X} \in \mathrm{Gr}_{n,p}$, let $\exp([\mathrm{Log}_{\mathbf{I}_{n,p}}^{gr}(\mathbf{X}), \mathbf{I}_{n,p}])\widetilde{\mathbf{I}}_{n,p} = \mathbf{V}\mathbf{U}$ be a QR decomposition, where $\widetilde{\mathbf{I}}_{n,p} = \begin{bmatrix} \mathbf{I}_p \\ 0 \end{bmatrix} \in \mathrm{M}_{n,p}$, $\mathbf{V} \in \mathrm{M}_{n,p}$ is a matrix with orthonormal columns, and $\mathbf{U} \in \mathrm{M}_{p,p}$ is an upper-triangular matrix. Then the non-linear activation function (Nair & Hinton, 2010; Huang et al., 2018) is given by

$$\sigma(\mathbf{X}) = \mathbf{V}\mathbf{V}^T.$$

Let $\mathbf{B}^l \in \mathrm{Gr}_{n,p}$ be the bias for layer $l$. Then the message-passing update of our network can be summarized as

$$\mathbf{P}_i^l = \phi_{\mathbf{M}^l}(\mathbf{X}_i^{l-1}) \qquad \text{(feature transformation)}$$

$$\mathbf{Q}_i^l = \mathrm{Exp}_{\mathbf{I}_{n,p}}^{gr}\Big( \sum_{j \in \mathcal{N}(i)} k_{i,j} \, \mathrm{Log}_{\mathbf{I}_{n,p}}^{gr}(\mathbf{P}_j^l) \Big) \quad \text{(aggregation)}$$

$$\mathbf{X}_i^l = \sigma(\mathbf{B}^l \oplus_{gr} \mathbf{Q}_i^l) \qquad \text{(bias and nonlinearity)}$$

The Grassmann logarithmic maps in the aggregation operation are obtained using Proposition 3.12.

Another approach for embedding graphs on Grassmann manifolds has also been proposed in Zhou et al. (2022). However, unlike our method, this method creates a Grassmann representation for a graph via a SVD of the matrix formed from node embeddings previously learned by a Euclidean neural network. Therefore, it is not designed to learn node embeddings on Grassmann manifolds.

## 4 EXPERIMENTS

### 4.1 HUMAN ACTION RECOGNITION

We use three datasets, i.e., HDM05 (Müller et al., 2007), FPHA (Garcia-Hernando et al., 2018), and NTU RBG+D 60 (NTU60) (Shahroudy et al., 2016). We compare our networks against the following state-of-the-art models: SPDNet (Huang & Gool, 2017)[1], SPDNetBN (Brooks et al., 2019)[2], SPSD-AI (Nguyen, 2022a), GyroAI-HAUNet (Nguyen, 2022b), and MLR-AI (Nguyen & Yang, 2023).

### 4.1.1 ABLATION STUDY

**Convolutional layers in SPD neural networks**  Our network GyroSpd++ has a MLR layer stacked on top of a convolutional layer (see Fig. 1). The motivation for using a convolutional layer

---

[1] https://github.com/zhiwu-huang/SPDNet.
[2] https://papers.nips.cc/paper/2019/hash/6e69ebbfad976d4637bb4b39de261bf7-Abstract.html.

Table 1: Results (mean accuracy $\pm$ standard deviation) and model sizes (MB) of various SPD neural networks on the three datasets (computed over 5 runs).

| Method | HDM05 | #HDM05 | FPHA | #FPHA | NTU60 | #NTU60 |
|---|---|---|---|---|---|---|
| SPDNet | $71.36 \pm 1.49$ | 6.58 | $88.79 \pm 0.36$ | 0.99 | $76.14 \pm 1.43$ | 1.80 |
| SPDNetBN | $75.05 \pm 1.38$ | 6.68 | $91.02 \pm 0.25$ | 1.03 | $78.35 \pm 1.34$ | 2.06 |
| GyroAI-HAUNet | $77.05 \pm 1.35$ | 0.31 | $95.65 \pm 0.23$ | 0.11 | $93.27 \pm 1.29$ | 0.02 |
| SPSD-AI | $79.64 \pm 1.54$ | 0.31 | $95.72 \pm 0.44$ | 0.11 | $93.92 \pm 1.55$ | 0.03 |
| MLR-AI | $78.26 \pm 1.37$ | 0.60 | $95.70 \pm 0.26$ | 0.21 | $94.27 \pm 1.32$ | 0.05 |
| GyroSpd++ (Ours) | $\mathbf{79.78} \pm 1.42$ | 0.76 | $96.84 \pm 0.27$ | 0.27 | $95.28 \pm 1.37$ | 0.07 |
| GyroSpsd++ (Ours) | $78.52 \pm 1.34$ | 0.75 | $\mathbf{97.90} \pm 0.24$ | 0.27 | $\mathbf{96.64} \pm 1.35$ | 0.07 |

Table 2: Results and computation times (seconds) per epoch of Gr-GCN++ and its variant based on the ONB perspective. Node embeddings are learned on $\widetilde{\mathrm{Gr}}_{14,7}$ and $\mathrm{Gr}_{14,7}$ for Gr-GCN-ONB and Gr-GCN++, respectively. Results are computed over 5 runs.

| Method | | Gr-GCN-ONB | Gr-GCN++ |
|---|---|---|---|
| Airport | Accuracy $\pm$ standard deviation | $81.9 \pm 1.2$ | $\mathbf{82.8} \pm 0.7$ |
| | Training | 0.49 | 0.97 |
| | Testing | 0.40 | 0.69 |
| Pubmed | Accuracy $\pm$ standard deviation | $76.2 \pm 1.5$ | $\mathbf{80.3} \pm 0.5$ |
| | Training | 3.40 | 6.48 |
| | Testing | 2.76 | 4.47 |
| Cora | Accuracy $\pm$ standard deviation | $68.1 \pm 1.0$ | $\mathbf{81.6} \pm 0.4$ |
| | Training | 0.57 | 0.77 |
| | Testing | 0.46 | 0.52 |

is that it can extract global features from local ones (covariance matrices computed from joint coordinates within sub-sequences of an action sequence). We use Affine-Invariant metrics for the convolutional layer and Log-Euclidean metrics for the MLR layer. Results in Tab. 1 show that GyroSpd++ consistently outperforms the SPD baselines in terms of mean accuracy. Results of GyroSpd++ with different designs of Riemannian metrics for its layers are given in Appendix D.4.1.

**MLR in structure spaces** We build GyroSpsd++ by replacing the MLR layer of GyroSpd++ with a MLR layer proposed in Section 3.3. Results of GyroSpsd++ are given in Tab. 1. Except SPSD-AI, GyroSpsd++ outperforms the other baselines on HDM05 dataset in terms of mean accuracy. Furthermore, GyroSpsd++ outperforms GyroSpd++ and all the baselines on FPHA and NTU60 datasets in terms of mean accuracy. These results show that MLR is effective when being designed in structure spaces from a gyrovector space perspective.

## 4.2 NODE CLASSIFICATION

We use three datasets, i.e., Airport (Zhang & Chen, 2018), Pubmed (Namata et al., 2012a), and Cora (Sen et al., 2008), each of them contains a single graph with thousands of labeled nodes. We compare our network Gr-GCN++ (see Fig. 1) against its variant Gr-GCN-ONB (see Appendix E.2.4) based on the ONB perspective. Results are shown in Tab. 2. Both networks give the best performance for $n = 14$ and $p = 7$. It can be seen that Gr-GCN++ outperforms Gr-GCN-ONB in all cases. The performance gaps are significant on Pubmed and Cora datasets.

## 5 CONCLUSION

In this paper, we develop FC and convolutional layers for SPD neural networks, and MLR on SPSD manifolds. We show how to perform backpropagation with the Grassmann logarithmic map in the projector perspective. Based on this method, we extend GCNs to Grassmann geometry. Finally, we present our experimental results demonstrating the efficacy of our approach in the human action recognition and node classification tasks.

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

| Symbol | Name |
|---|---|
| $\mathrm{M}_{n,m}$ | Space of $n \times m$ matrices |
| $\mathrm{Sym}_n^+$ | Space of $n \times n$ SPD matrices |
| $\mathrm{Sym}_n^{+,ai}$ | Space of $n \times n$ SPD matrices with AI geometry |
| $\mathrm{Sym}_n$ | Space of $n \times n$ symmetric matrices |
| $\mathrm{Gr}_{n,p}$ | Grassmannian in the projector perspective |
| $\widetilde{\mathrm{Gr}}_{n,p}$ | Grassmannian in the ONB perspective |
| $\mathrm{S}_{n,p}^+$ | Space of $n \times n$ SPSD matrices of rank $p \leq n$ |
| $\mathcal{M}$ | Matrix manifold |
| $T_{\mathbf{P}}\mathcal{M}$ | Tangent space of $\mathcal{M}$ at $\mathbf{P}$ |
| $\exp(\mathbf{P})$ | Matrix exponential of $\mathbf{P}$ |
| $\log(\mathbf{P})$ | Matrix logarithm of $\mathbf{P}$ |
| $\mathrm{Exp}_{\mathbf{P}}^{ai}(\mathbf{W})$ | Exponential map of $\mathbf{W}$ at $\mathbf{P}$ in $\mathrm{Sym}_n^{+,ai}$ |
| $\mathrm{Log}_{\mathbf{P}}^{ai}(\mathbf{Q})$ | Logarithmic map of $\mathbf{Q}$ at $\mathbf{P}$ in $\mathrm{Sym}_n^{+,ai}$ |
| $\mathcal{T}_{\mathbf{P}\to\mathbf{Q}}^{ai}(\mathbf{W})$ | Parallel transport of $\mathbf{W}$ from $\mathbf{P}$ to $\mathbf{Q}$ in $\mathrm{Sym}_n^{+,ai}$ |
| $\mathrm{Exp}_{\mathbf{P}}^{gr}(\mathbf{W})$ | Exponential map of $\mathbf{W}$ at $\mathbf{P}$ in $\mathrm{Gr}_{n,p}$ |
| $\mathrm{Log}_{\mathbf{P}}^{gr}(\mathbf{Q})$ | Logarithmic map of $\mathbf{Q}$ at $\mathbf{P}$ in $\mathrm{Gr}_{n,p}$ |
| $\widetilde{\mathrm{Log}}_{\mathbf{P}}^{gr}(\mathbf{Q})$ | Logarithmic map of $\mathbf{Q}$ at $\mathbf{P}$ in $\widetilde{\mathrm{Gr}}_{n,p}$ |
| $\oplus_{ai}, \ominus_{ai}$ | Binary and inverse operations in $\mathrm{Sym}_n^{+,ai}$ |
| $\oplus_{gr}, \ominus_{gr}$ | Binary and inverse operations in $\mathrm{Gr}_{n,p}$ |
| $\widetilde{\oplus}_{gr}, \widetilde{\ominus}_{gr}$ | Binary and inverse operations in $\widetilde{\mathrm{Gr}}_{n,p}$ |
| $\oplus_{psd,ai}$ | Binary operation in $\widetilde{\mathrm{Gr}}_{n,p} \times \mathrm{Sym}_p^{+,ai}$ |
| $\ominus_{psd,ai}$ | Inverse operation in $\widetilde{\mathrm{Gr}}_{n,p} \times \mathrm{Sym}_p^{+,ai}$ |
| $\mathcal{H}_{\mathbf{W},\mathbf{P}}^{spd,ai}$ | Hypergyroplane in $\mathrm{Sym}_n^{+,ai}$ |
| $\mathcal{H}_{\mathbf{W},\mathbf{P}}^{psd,ai}$ | Hypergyroplane in $\widetilde{\mathrm{Gr}}_{n,p} \times \mathrm{Sym}_p^{+,ai}$ |
| $E_{(i,j)}^{ai}$ | Orthonormal basis of $\mathrm{Sym}_m^{+,ai}$ |
| $\langle.,.\rangle^{ai}$ | Inner product in $\mathrm{Sym}_n^{+,ai}$ |
| $\langle.,.\rangle^{gr}$ | Inner product in $\mathrm{Gr}_{n,p}$ |
| $\langle.,.\rangle^{psd,ai}$ | Inner product in $\widetilde{\mathrm{Gr}}_{n,p} \times \mathrm{Sym}_p^{+,ai}$ |
| $\langle.,.\rangle_{\mathbf{P}}^{ai}$ | Affine-Invariant metric at $\mathbf{P}$ |
| $\mathbf{I}_n$ | $n \times n$ identity matrix |
| $\mathbf{I}_{n,p}$ | $\begin{bmatrix} \mathbf{I}_p & 0 \\ 0 & 0 \end{bmatrix} \in \mathrm{M}_{n,n}$ |
| $\widetilde{\mathbf{I}}_{n,p}$ | $\begin{bmatrix} \mathbf{I}_p \\ 0 \end{bmatrix} \in \mathrm{M}_{n,p}$ |

Table 3: The main notations used in the paper. For the notations related to SPD manifolds, only those associated with Affine-Invariant geometry are shown.

# A  NOTATIONS

Tab. 3 presents the main notations used in our paper.

# B  MLR IN STRUCTURE SPACES

Algorithm 1 summarizes all steps for the computation of pseudo-gyrodistances in Theorem 3.11.

Details of some steps are given below:

---

**Algorithm 1:** Computation of Pseudo-gyrodistances

---

**Input:** A batch of SPSD matrices $\mathbf{X}_i \in \mathrm{S}_{n,p}^+, i = 1, \ldots, N$
 The number of classes $C$
 The parameters for each class $\mathbf{U}_P^c, \mathbf{U}_W^c \in \widetilde{\mathrm{Gr}}_{n,p}, \mathbf{S}_P^c, \mathbf{S}_W^c \in \mathrm{Sym}_p^+, c = 1, \ldots, C$
 A constant $\gamma \in [0, 1]$
**Output:** An array $d \in \mathrm{M}_{N,C}$ of pseudo-gyrodistances

1   $\mathbf{U}^m \leftarrow \widetilde{\mathbf{I}}_{n,p}$;
2   $(\mathbf{U}_i, \boldsymbol{\Sigma}_i, \mathbf{V}_i)_{i=1,\ldots,N} \leftarrow \mathrm{SVD}((\mathbf{X}_i)_{i=1,\ldots,N})$;               /$\star$   $\mathbf{X}_i = \mathbf{U}_i\boldsymbol{\Sigma}_i\mathbf{V}_i^T$   $\star$/
3   $(\mathbf{U}_i)_{i=1,\ldots,N} \leftarrow (\mathbf{U}_i[:, :p])_{i=1,\ldots,N}$;
4   **if** *training* **then**
5     $\mathbf{U} \leftarrow \mathrm{GrMean}((\mathbf{U}_i)_{i=1,\ldots,N})$;
6     $\mathbf{U}^m \leftarrow \mathrm{GrGeodesic}(\mathbf{U}^m, \mathbf{U}, \gamma)$;
7   **end**
8   $(\mathbf{U}_X^i, \mathbf{S}_X^i)_{i=1,\ldots,N} = \mathrm{GrCanonicalize}((\mathbf{X}_i, \mathbf{U}_i)_{i=1,\ldots,N}, \mathbf{U}^m)$;
9   **for** $c \leftarrow 1$ **to** $C$ **do**
10     $d((\mathbf{X})_{i=1,\ldots,N}, c) = \dfrac{|\lambda\langle(\widetilde{\ominus}_{gr}\mathbf{U}_P^c\widetilde{\oplus}_{gr}\mathbf{U}_X)(\widetilde{\ominus}_{gr}\mathbf{U}_P^c\widetilde{\oplus}_{gr}\mathbf{U}_X)^T, \mathbf{U}_W^c(\mathbf{U}_W^c)^T\rangle^{gr} + \langle\ominus_g\mathbf{S}_P^c\oplus_g\mathbf{S}_X, \mathbf{S}_W^c\rangle^g|}{\sqrt{\lambda(\|\mathbf{U}_W^c(\mathbf{U}_W^c)^T\|^{gr})^2 + (\|\mathbf{S}_W^c\|^g)^2}}$;
11   **end**

---

- $\mathrm{SVD}((\mathbf{X}_i)_{i=1,\ldots,N})$ performs singular value decompositions for a batch of matrices.
- $\mathrm{GrMean}((\mathbf{U}_i)_{i=1,\ldots,N})$ computes the Fréchet mean of its arguments.
- $\mathrm{GrGeodesic}(\mathbf{U}^m, \mathbf{U}, \gamma)$ computes a point on a geodesic from $\mathbf{U}^m$ to $\mathbf{U}$ at step $\gamma$ ($\gamma = 0.1$ in our experiments).
- $\mathrm{GrCanonicalize}((\mathbf{X}_i, \mathbf{U}_i)_{i=1,\ldots,N}, \mathbf{V})$ computes the canonical representations of $\mathbf{X}_i, i = 1, \ldots, N$ using $\mathbf{V}$ as a common subspace (see Appendix H).
- Line 10: $\mathbf{U}_X = (\mathbf{U}_X^i)_{i=1,\ldots,N}, \mathbf{S}_X = (\mathbf{S}_X^i)_{i=1,\ldots,N}$, and the computation of pseudo-gyrodistances is performed in batches.

## C   FORMULATION OF MLR FROM THE PERSPECTIVE OF DISTANCES TO HYPERPLANES

Given $K$ classes, MLR computes the probability of each of the output classes as

$$p(y = k|x) = \frac{\exp(w_k^T x + b_k)}{\sum_{i=1}^K \exp(w_i^T x + b_i)} \propto \exp(w_k^T x + b_k), \tag{3}$$

where $x$ is an input sample, $b_i \in \mathbb{R}, x, w_i \in \mathbb{R}^n, i = 1, \ldots, K$.

As shown in Lebanon & Lafferty (2004), Eq. (3) can be rewritten as

$$p(y = k|x) \propto \exp(\mathrm{sign}(w_k^T x + b_k)\|w_k\|d(x, \mathcal{H}_{w_k, b_k})),$$

where $d(x, \mathcal{H}_{w_k, b_k})$ is the distance from point $x$ to a hyperplane $\mathcal{H}_{w_k, b_k}$ defined as

$$\mathcal{H}_{w,b} = \{x \in \mathbb{R}^n : \langle w, x\rangle + b = 0\},$$

where $w \in \mathbb{R}^n \setminus \{\mathbf{0}\}$, and $b \in \mathbb{R}$.

## D   HUMAN ACTION RECOGNITION

### D.1   DATASETS

**HDM05 (Müller et al., 2007)**   It has 2337 sequences of 3D skeleton data classified into 130 classes. Each frame contains the 3D coordinates of 31 body joints. We use all the action classes and follow the experimental protocol in Harandi et al. (2018) in which 2 subjects are used for training and the remaining 3 subjects are used for testing.

**FPHA (Garcia-Hernando et al., 2018)**   It has 1175 sequences of 3D skeleton data classified into 45 classes. Each frame contains the 3D coordinates of 21 hand joints. We follow the experimental protocol in Garcia-Hernando et al. (2018) in which 600 sequences are used for training and 575 sequences are used for testing.

**NTU60 (Shahroudy et al., 2016)**   It has 56880 sequences of 3D skeleton data classified into 60 classes. Each frame contains the 3D coordinates of 25 or 50 body joints. We use the mutual actions and follow the cross-subject experimental protocol in Shahroudy et al. (2016) in which data from 20 subjects are used for training, and those from the other 20 subjects are used for testing.

## D.2   IMPLEMENTATION DETAILS

### D.2.1   SETUP

We use the PyTorch framework to implement our networks and those from previous works. These networks are trained using cross-entropy loss and Adadelta optimizer for 2000 epochs. The learning rate is set to $10^{-3}$. The factors $\beta$ (see Proposition 3.4) and $\lambda$ (see Definition 3.9) are set to 0 and 1, respectively. For GyroSpd++, the sizes of output matrices of the convolutional layer are set to $34 \times 34$, $21 \times 21$, and $11 \times 11$ for the experiments on HDM05, FPHA, and NTU60 datasets, respectively. For GyroSpsd++, the sizes of SPD matrices in structure spaces are set to $20 \times 20$, $14 \times 14$, and $8 \times 8$ for the experiments on HDM05, FPHA, and NTU60 datasets, respectively. We use a batch size of 32 for HDM05 and FPHA datasets, and a batch size of 256 for NTU60 dataset.

### D.2.2   INPUT DATA

We use a similar method as in Nguyen (2022b) to compute the input data for our network GyroSpd++. We first identify a closest left (right) neighbor of every joint based on their distance to the hip (wrist) joint, and then combine the 3D coordinates of each joint and those of its left (right) neighbor to create a feature vector for the joint. For a given frame $t$, a mean vector $\boldsymbol{\mu}_t$ and a covariance matrix $\boldsymbol{\Sigma}_t$ are computed from the set of feature vectors of the frame and then combined to create an SPD matrix as

$$\mathbf{Y}_t = \begin{bmatrix} \boldsymbol{\Sigma}_t + \boldsymbol{\mu}_t(\boldsymbol{\mu}_t)^T & \boldsymbol{\mu}_t \\ (\boldsymbol{\mu}_t)^T & 1 \end{bmatrix}.$$

The lower part of matrix $\log(\mathbf{Y}_t)$ is flattened to obtain a vector $\tilde{\mathbf{v}}_t$. All vectors $\tilde{\mathbf{v}}_t$ within a time window $[t, t+c-1]$, where $c$ is determined from a temporal pyramid representation of the sequence (the number of temporal pyramids is set to 2 in our experiments), are used to compute a covariance matrix as

$$\mathbf{Z}_t = \frac{1}{c} \sum_{i=t}^{t+c-1} (\tilde{\mathbf{v}}_i - \bar{\mathbf{v}}_t)(\tilde{\mathbf{v}}_i - \bar{\mathbf{v}}_t)^T, \tag{4}$$

where $\bar{\mathbf{v}}_t = \frac{1}{c} \sum_{i=t}^{t+c-1} \tilde{\mathbf{v}}_i$. For GyroAI-HAUNet, SPSD-AI, MLR-AI, GyroSpd++, and GyroSpsd++, the input data are the set of matrices obtained in Eq. (4).

For SPDNet and SPDNetBN, each sequence is represented by a covariance matrix (Huang & Gool, 2017; Brooks et al., 2019). The sizes of the covariance matrices are $93 \times 93$, $60 \times 60$, and $150 \times 150$ for HDM05, FPHA, and NTU60 datasets, respectively. For SPDNet, the same architecture as the one in Huang & Gool (2017) is used with three Bimap layers. For SPDNetBN, the same architecture as the one in Brooks et al. (2019) is used with three Bimap layers. The sizes of the transformation matrices for the experiments on HDM05, FPHA, and NTU60 datasets are set to $93 \times 93$, $60 \times 60$, and $150 \times 150$, respectively.

### D.2.3   CONVOLUTIONAL LAYERS

In order to reduce the number of parameters and the computational cost for the convolutional layer in GyroSpd++, we assume a diagonal structure for the parameter $\mathbf{P}_{(i,j)}$ (see Propositions 3.4, 3.5, and 3.6), i.e.,

$$\mathbf{P}_{(i,j)} = \mathrm{concat}_{spd}(\mathbf{P}_{(i,j)}^1, \ldots, \mathbf{P}_{(i,j)}^L),$$

where $L$ is the number of input SPD matrices of operation $\mathrm{concat}_{spd}(.)$.

### D.2.4 Optimization

For parameters that are SPD matrices, we model them on the space of symmetric matrices, and then apply the exponential map at the identity.

For any parameter $\mathbf{P} \in \widetilde{\mathrm{Gr}}_{n,p}$, we parameterize it by a matrix $\mathbf{B} \in \mathrm{M}_{p,n-p}$ such that

$$\begin{bmatrix} 0 & \mathbf{B} \\ -\mathbf{B}^T & 0 \end{bmatrix} = [\mathrm{Log}^{gr}_{\mathbf{I}_{n,p}}(\mathbf{PP}^T), \mathbf{I}_{n,p}].$$

Then parameter $\mathbf{P}$ can be computed by

$$\mathbf{P} = \exp([\mathrm{Log}^{gr}_{\mathbf{I}_{n,p}}(\mathbf{PP}^T), \mathbf{I}_{n,p}])\widetilde{\mathbf{I}}_{n,p} = \exp\left(\begin{bmatrix} 0 & \mathbf{B} \\ -\mathbf{B}^T & 0 \end{bmatrix}\right)\widetilde{\mathbf{I}}_{n,p}.$$

Thus, we can optimize all parameters on Euclidean spaces without having to resort to techniques developed on Riemannian manifolds.

### D.3 Time complexity analysis

Let $n_{in} \times n_{in}$ be the size of input SPD matrices, $n_{out} \times n_{out}$ be the size of output matrices of the convolutional layer in GyroSpd++, $n_{rank} \times n_{rank}$ be the size of SPD matrices in structure spaces, $n_c$ be the number of action classes, $n_s$ be the number of SPD matrices encoding a sequence.

- GyroSpd++: The convolutional layer has time complexity $O(n_s n_{out}^2 n_{in}^3)$. The MLR layer has time complexity $O(n_c n_{out}^3)$.
- GyroSpsd++: The convolutional layer has time complexity $O(n_s n_{out}^2 n_{in}^3)$. The MLR layer has time complexity $O(n_{out}^3 + n_c n_{rank}^3)$.

### D.4 More Experimental Results

#### D.4.1 Ablation Study

**Impact of the factor $\beta$ in Affine-Invariant metrics**    To study the impact of the factor $\beta$ in Affine-Invariant metrics on the performance of GyroSpd++, we follow the approach in Nguyen (2021). Denote by $(\boldsymbol{\mu}, \boldsymbol{\Sigma})$ the Gaussian distribution where $\boldsymbol{\mu} \in \mathbb{R}^n$ and $\boldsymbol{\Sigma} \in \mathrm{M}_{n,n}$ are its mean and covariance. We can identify the Gaussian distribution $(\boldsymbol{\mu}, \boldsymbol{\Sigma})$ with the following matrix:

$$(\det \boldsymbol{\Sigma})^{-\frac{1}{n+k}} \begin{bmatrix} \boldsymbol{\Sigma} + k\boldsymbol{\mu}\boldsymbol{\mu}^T & \boldsymbol{\mu}(k) \\ \boldsymbol{\mu}(k)^T & \mathbf{I}_k \end{bmatrix},$$

where $k \geq 1$, $\boldsymbol{\mu}(k)$ is a matrix with $k$ identical column vectors $\boldsymbol{\mu}$. The natural symmetric Riemannian metric resulting from the above embedding is given (Nguyen, 2021) by

$$\langle \mathbf{V}, \mathbf{W} \rangle_{\mathbf{P}} = \mathrm{Tr}(\mathbf{V}\mathbf{P}^{-1}\mathbf{W}\mathbf{P}^{-1}) - \frac{1}{n+k} \mathrm{Tr}(\mathbf{V}\mathbf{P}^{-1}) \mathrm{Tr}(\mathbf{W}\mathbf{P}^{-1}),$$

where $\mathbf{P}$ is an SPD matrix, $\mathbf{V}$ and $\mathbf{W}$ are two tangent vectors at point $\mathbf{P}$ of the manifold. This Riemannian metric belongs to the family of Affine-Invariant metrics where $\beta = -\frac{1}{n+k} > -\frac{1}{n}$.

For this study, we replace matrix $\mathbf{Z}_t$ in Eq. (4) with the following matrix:

$$\widetilde{\mathbf{Z}}_t = (\det \mathbf{Z}_t)^{-\frac{1}{n_1+k}} \begin{bmatrix} \mathbf{Z}_t + k\bar{\mathbf{v}}_t\bar{\mathbf{v}}_t^T & \bar{\mathbf{v}}_t(k) \\ \bar{\mathbf{v}}_t(k)^T & \mathbf{I}_k \end{bmatrix},$$

where $n_1 \times n_1$ is the size of matrix $\mathbf{Z}_t$. The input data for GyroSpd++ are then computed from $\widetilde{\mathbf{Z}}_t$ as before.

Tab. 4 reports the mean accuracies and standard deviations of GyroSpd++ with respect to different settings of $\beta$ on the three datasets. GyroSpd++ with the setting $\beta = 0$ generally works well on all the datasets. Setting $k = 3$ improves the accuracy of GyroSpd on NTU60 dataset. We also observe that setting $k$ to a high value, e.g., $k = 10$ lowers the accuracies of GyroSpd++ on the datasets.

Table 4: Results (mean accuracy $\pm$ standard deviation) of GyroSpd++ with respect to different settings of $\beta$ on the three datasets (computed over 5 runs).

| Dataset | HDM05 | FPHA | NTU60 |
|---|---|---|---|
| $\beta = 0$ | **79.78** $\pm$ 1.42 | **96.84** $\pm$ 0.27 | 95.28 $\pm$ 1.37 |
| $k = 3$ | 79.12 $\pm$ 1.37 | 96.16 $\pm$ 0.25 | **96.32** $\pm$ 1.33 |
| $k = 10$ | 78.25 $\pm$ 1.39 | 95.91 $\pm$ 0.29 | 94.44 $\pm$ 1.34 |

Table 5: Results and computation times (seconds) of GyroSpd++ with respect to different settings of the output dimension of the convolutional layer on FPHA dataset (computed over 5 runs). Experiments are conducted on a machine with Intel Core i7-8565U CPU 1.80 GHz 24GB RAM.

| $m$ | Accuracy $\pm$ standard deviation | Computation time/epoch | |
|---|---|---|---|
| | | Training | Testing |
| 10 | 94.53 $\pm$ 0.31 | 30.08 | 12.07 |
| 21 | **96.84** $\pm$ 0.27 | 129.30 | 50.84 |
| 30 | 96.80 $\pm$ 0.26 | 182.52 | 71.49 |

**Output dimension of convolutional layers** Tab. 5 presents results and computation times of GyroSpd++ with respect to different settings of the output dimension of the convolutional layer on FPHA dataset. Results show that the setting $m = 21$ clearly outperforms the setting $m = 10$ in terms of mean accuracy and standard deviation. However, compared to the setting $m = 21$, the setting $m = 30$ only increases the training and testing times without improving the mean accuracy of GyroSpd++.

**Design of Riemannian metrics for network blocks** The use of different Riemannian metrics for the convolutional and MLR layers of GyroSpd++ results in different variants of the same architecture. Results of some of these variants on FPHA dataset are shown in Tab. 6. It is noted that our architecture gives the best performance in terms of mean accuracy, while the architecture with Log-Cholesky geometry for the MLR layer performs the worst in terms of mean accuracy.

D.4.2 COMPARISON OF GYROSPD++ AGAINST STATE-OF-THE-ART METHODS

Here we present more comparisons of our networks against state-of-the-art networks. These networks belong to one of the following families of neural networks: **(1) Hyperbolic neural networks**: HypGRU (Ganea et al., 2018)[3]; **(2) Graph neural networks**: MS-G3D (Liu et al., 2020)[4], TGN (Zhou et al., 2023)[5]; **(3) Transformers**: ST-TR (Plizzari et al., 2021)[6]. Note that MS-G3D, TGN, and ST-TR are specifically designed for skeleton-based action recognition. We use default parameter settings for these networks. Results of our networks and their competitors on HDM05, FPHA, and NTU60 datasets are shown in Tabs. 7, 8, and 9, respectively. On HDM05 dataset, GyroSpd++ outperforms HypGRU, MS-G3D, ST-TR, and TGN by 25.6%, 10.8%, 11.9%, and 9.5% points in terms of mean accuracy, respectively. On FPHA dataset, GyroSpd++ outperforms HypGRU, MS-G3D, ST-TR, and TGN by 38.6%, 8.5%, 10.9%, and 6.0% points in terms of mean accuracy, respectively. On NTU60 dataset, GyroSpd++ outperforms HypGRU, MS-G3D, ST-TR, and TGN by 7.0%, 3.1%, 4.2%, and 2.2% points in terms of mean accuracy, respectively. Overall, our networks are superior to their competitors in all cases.

Finally, we present a comparison of computation times of SPD neural networks in Tab. 10.

Table 6: Results (mean accuracy $\pm$ standard deviation) of GyroSpd++ with different designs of Riemannian metrics for its layers on FPHA dataset (computed over 5 runs).

| AI-LE (GyroSpd++) | LE-LE | AI-AI | LE-AI | AI-LC |
|---|---|---|---|---|
| **96.84** $\pm$ 0.27 | 94.72 $\pm$ 0.25 | 94.35 $\pm$ 0.29 | 95.21 $\pm$ 0.26 | 89.16 $\pm$ 0.26 |

Table 7: Results of our networks and some state-of-the-art methods on HDM05 dataset (computed over 5 runs).

| Method | Accuracy $\pm$ Standard deviation |
|---|---|
| HypGRU (Ganea et al., 2018) | 54.18 $\pm$ 1.51 |
| MS-G3D (Liu et al., 2020) | 68.92 $\pm$ 1.72 |
| ST-TR (Plizzari et al., 2021) | 67.84 $\pm$ 1.66 |
| TGN (Zhou et al., 2023) | 70.26 $\pm$ 1.48 |
| GyroSpd++ (Ours) | **79.78** $\pm$ 1.42 |
| GyroSpsd++ (Ours) | 78.52 $\pm$ 1.34 |

# E  NODE CLASSIFICATION

## E.1  DATASETS

**Airport (Chami et al., 2019)**   It is a flight network dataset from OpenFlights.org where nodes represent airports, edges represent the airline Routes, and node labels are the populations of the country where the airport belongs.

**Pubmed (Namata et al., 2012b)**   It is a standard benchmark describing citation networks where nodes represent scientific papers in the area of medicine, edges are citations between them, and node labels are academic (sub)areas.

**Cora (Sen et al., 2008)**   It is a citation network where nodes represent scientific papers in the area of machine learning, edges are citations between them, and node labels are academic (sub)areas.

The statistics of the three datasets are summarized in Tab. 11.

## E.2  IMPLEMENTATION DETAILS

### E.2.1  SETUP

Our network is implemented using the PyTorch framework. We set hyperparameters as in Zhao et al. (2023) that are found via grid search for each graph architecture on the development set of a given dataset. The best settings of $n$ and $p$ are found from $(n, p) \in \{(2k, k)\}, k = 2, 3, \ldots, 10$. The batch size is set to the total number of graph nodes in a dataset (Chami et al., 2019; Zhao et al., 2023). The networks are trained using cross-entropy loss and Adam optimizer for a maximum of 500 epochs. The learning rate is set to $10^{-2}$. Early stopping is used when the loss on the development set has not decreased for 200 epochs. Each network has two layers that perform message passing twice at one iteration (Zhao et al., 2023). We use the 70/15/15 percent splits (Chami et al., 2019) for Airport dataset, and standard splits in GCN Kipf & Welling (2017) for Pubmed and Cora datasets.

---

[3] `https://github.com/dalab/hyperbolic_nn`.
[4] `https://github.com/kenziyuliu/MS-G3D`.
[5] `https://github.com/zhysora/FR-Head`.
[6] `https://github.com/Chiaraplizz/ST-TR`.

Table 8: Results of our networks and some state-of-the-art methods on FPHA dataset (computed over 5 runs).

| Method | Accuracy ± Standard deviation |
|---|---|
| HypGRU (Ganea et al., 2018) | $58.24 \pm 0.29$ |
| MS-G3D (Liu et al., 2020) | $88.26 \pm 0.67$ |
| ST-TR (Plizzari et al., 2021) | $85.94 \pm 0.46$ |
| TGN (Zhou et al., 2023) | $90.81 \pm 0.53$ |
| GyroSpd++ (Ours) | $96.84 \pm 0.27$ |
| GyroSpsd++ (Ours) | $\mathbf{97.90} \pm 0.24$ |

Table 9: Results of our networks and some state-of-the-art methods on NTU60 dataset (computed over 5 runs).

| Method | Accuracy ± Standard deviation |
|---|---|
| HypGRU (Ganea et al., 2018) | $88.26 \pm 1.40$ |
| MS-G3D (Liu et al., 2020) | $92.15 \pm 1.60$ |
| ST-TR (Plizzari et al., 2021) | $91.04 \pm 1.52$ |
| TGN (Zhou et al., 2023) | $93.02 \pm 1.56$ |
| GyroSpd++ (Ours) | $95.28 \pm 1.37$ |
| GyroSpsd++ (Ours) | $\mathbf{96.64} \pm 1.35$ |

### E.2.2 GRASSMANN LOGARITHMIC MAP IN THE ONB PERSPECTIVE

The Grassmann logarithmic map in the ONB perspective is given (Edelman et al., 1998) by

$$\widetilde{\mathrm{Log}}^{gr}_{\mathbf{P}}(\mathbf{Q}) = \mathbf{U}\arctan(\mathbf{\Sigma})\mathbf{V}^T,$$

where $\mathbf{P}, \mathbf{Q} \in \widetilde{\mathrm{Gr}}_{n,p}$, $\mathbf{U}, \mathbf{\Sigma}$, and $\mathbf{V}$ are obtained from the SVD $(\mathbf{I}_n - \mathbf{P}\mathbf{P}^T)\mathbf{Q}(\mathbf{P}^T\mathbf{Q})^{-1} = \mathbf{U}\mathbf{\Sigma}\mathbf{V}^T$.

### E.2.3 GR-GCN++

To create Grassmann embeddings as input node features, we first transform $d$-dimensional input features into $p(n - p)$-dimensional vectors via a linear map. We then reshape each resulting vector to a matrix $\mathbf{B} \in \mathrm{M}_{p,n-p}$. The input Grassmann embedding $\mathbf{X}^0_i, i \in \mathcal{V}$ is computed as

$$\mathbf{X}^0_i = \exp\left(\begin{bmatrix} 0 & \mathbf{B} \\ -\mathbf{B}^T & 0 \end{bmatrix}\right)\mathbf{I}_{n,p}\exp\left(-\begin{bmatrix} 0 & \mathbf{B} \\ -\mathbf{B}^T & 0 \end{bmatrix}\right).$$

### E.2.4 GR-GCN-ONB

To create Grassmann embeddings as input node features, we first transform $d$-dimensional input features into $p(n - p)$-dimensional vectors via a linear map. We then reshape each resulting vector to a matrix $\mathbf{B} \in \mathrm{M}_{p,n-p}$. The input Grassmann embedding $\mathbf{X}^0_i, i \in \mathcal{V}$ is computed as

$$\mathbf{X}^0_i = \exp\left(\begin{bmatrix} 0 & \mathbf{B} \\ -\mathbf{B}^T & 0 \end{bmatrix}\right)\widetilde{\mathbf{I}}_{n,p}.$$

Feature transformation is performed by first mapping the input to a projection matrix (using the mapping $\tau$ in Section 3.4.1), then applying an isometry map based on left Grassmann gyrotranslations (Nguyen & Yang, 2023), and finally mapping the result back to a matrix with orthonormal columns. This is equivalent to performing the following mapping:

$$\phi_{\mathbf{M}}(\mathbf{X}^l_i) = \mathbf{M}\widetilde{\oplus}_{gr}\mathbf{X}^l_i = \exp([\mathrm{Log}^{gr}_{\mathbf{I}_{n,p}}(\mathbf{M}\mathbf{M}^T), \mathbf{I}_{n,p}])\mathbf{X}^l_i,$$

where $\mathbf{X}^l_i \in \widetilde{\mathrm{Gr}}_{n,p}$ and $\mathbf{M} \in \widetilde{\mathrm{Gr}}_{n,p}$ is a model parameter.

Table 10: Computation times (seconds) per epoch of our networks and some state-of-the-art SPD neural networks on FPHA dataset. Experiments are conducted on a machine with Intel Core i7-8565U CPU 1.80 GHz 24GB RAM.

| Method | SPDNet | SPDNetBN | GyroAI-HAUNet | SPSD-AI | MLR-AI | GyroSpd++ | GyroSpsd++ |
|--------|--------|----------|---------------|---------|--------|-----------|------------|
| Training | 17.52 | 40.08 | 62.21 | 73.73 | 102.58 | 129.30 | 126.08 |
| Testing | 3.48 | 6.22 | 30.83 | 35.54 | 46.28 | 50.84 | 48.06 |

Table 11: Description of the datasets for node classification.

| Dataset | #Nodes | #Edges | #Classes | #Features |
|---------|--------|--------|----------|-----------|
| Airport | 3188 | 18631 | 4 | 4 |
| Pubmed | 19717 | 44338 | 3 | 500 |
| Cora | 2708 | 5429 | 7 | 1433 |

For any $\mathbf{X} \in \widetilde{\mathrm{Gr}}_{n,p}$, let $\mathbf{X} = \mathbf{VU}$ be a QR decomposition of $\mathbf{X}$, where $\mathbf{V} \in \mathrm{M}_{n,p}$ is a matrix with orthonormal columns, and $\mathbf{U} \in \mathrm{M}_{p,p}$ is an upper-triangular matrix. Then the non-linear activation function is given by

$$\sigma(\mathbf{X}) = \mathbf{V}.$$

Bias addition is performed using operation $\widetilde{\oplus}_{gr}$ instead of operation $\oplus_{gr}$. The output of Gr-GCN-ONB is mapped to a projection matrix for node classification.

### E.2.5 Optimization

For any parameter $\mathbf{P} \in \mathrm{Gr}_{n,p}$, we parameterize it by a matrix $\mathbf{B} \in \mathrm{M}_{p,n-p}$ such that

$$\begin{bmatrix} 0 & \mathbf{B} \\ -\mathbf{B}^T & 0 \end{bmatrix} = [\mathrm{Log}^{gr}_{\mathbf{I}_{n,p}}(\mathbf{P}), \mathbf{I}_{n,p}].$$

Then parameter $\mathbf{P}$ can be computed by

$$\mathbf{P} = \exp\left( \begin{bmatrix} 0 & \mathbf{B} \\ -\mathbf{B}^T & 0 \end{bmatrix} \right) \mathbf{I}_{n,p} \exp\left( -\begin{bmatrix} 0 & \mathbf{B} \\ -\mathbf{B}^T & 0 \end{bmatrix} \right).$$

### E.3 More Experimental Results

#### E.3.1 Ablation Study

**Projector vs. ONB perspective**  More results of Gr-GCN++ and Gr-GCN-ONB are presented in Tabs. 12 and 13. As can be observed, Gr-GCN++ outperforms Gr-GCN-ONB in all cases. In particular, the former outperforms the latter by large margins on Airport and Cora datasets. Results show that while both the networks learn node embeddings on Grassmann manifolds, the choice of perspective for representing these embeddings and the associated parameters can have a significant impact on the network performance.

#### E.3.2 Comparison of Gr-GCN++ against State-of-the-Art Methods

Tab. 14 shows results of Gr-GCN++ and some state-of-the-art methods on the three datasets. The hyperbolic networks outperform their SPD and Grassmann counterparts on Airport dataset with high hyperbolicity (Chami et al., 2019). This agrees with previous works (Chami et al., 2019; Zhang et al., 2022) that report good performances of hyperbolic embeddings on tree-like datasets. However, our network and its SPD counterpart SPD-GCN outperform their competitors on Pubmed and Cora datasets with low hyperbolicities. Compared to SPD-GCN, Gr-GCN++ always gives more consistent results.

Table 12: Results and computation times (seconds) per epoch of Gr-GCN++ and its variant Gr-GCN-ONB based on the ONB perspective. Node embeddings are learned on $\widetilde{\mathrm{Gr}}_{4,2}$ and $\mathrm{Gr}_{4,2}$ for Gr-GCN-ONB and Gr-GCN++, respectively. Results are computed over 5 runs. Experiments are conducted on a machine with Intel Core i7-9700 CPU 3.00 GHz 15GB RAM.

| | Method | Gr-GCN-ONB | Gr-GCN++ |
|---|---|---|---|
| | Accuracy $\pm$ standard deviation | $53.2 \pm 1.9$ | $\mathbf{60.1 \pm 1.3}$ |
| Airport | Training | 0.07 | 0.21 |
| | Testing | 0.05 | 0.12 |
| | Accuracy $\pm$ standard deviation | $75.7 \pm 2.1$ | $\mathbf{77.5 \pm 1.1}$ |
| Pubmed | Training | 0.50 | 0.90 |
| | Testing | 0.38 | 0.54 |
| | Accuracy $\pm$ standard deviation | $33.9 \pm 2.3$ | $\mathbf{64.4 \pm 1.4}$ |
| Cora | Training | 0.10 | 0.12 |
| | Testing | 0.07 | 0.08 |

Table 13: Results and computation times (seconds) per epoch of Gr-GCN++ and its variant Gr-GCN-ONB based on the ONB perspective. Node embeddings are learned on $\widetilde{\mathrm{Gr}}_{6,3}$ and $\mathrm{Gr}_{6,3}$ for Gr-GCN-ONB and Gr-GCN++, respectively. Results are computed over 5 runs. Experiments are conducted on a machine with Intel Core i7-9700 CPU 3.00 GHz 15GB RAM.

| | Method | Gr-GCN-ONB | Gr-GCN++ |
|---|---|---|---|
| | Accuracy $\pm$ standard deviation | $65.8 \pm 1.5$ | $\mathbf{74.1 \pm 0.9}$ |
| Airport | Training | 0.19 | 0.34 |
| | Testing | 0.14 | 0.21 |
| | Accuracy $\pm$ standard deviation | $75.8 \pm 2.0$ | $\mathbf{78.5 \pm 0.9}$ |
| Pubmed | Training | 0.90 | 1.76 |
| | Testing | 0.75 | 1.05 |
| | Accuracy $\pm$ standard deviation | $41.4 \pm 2.2$ | $\mathbf{70.5 \pm 1.1}$ |
| Cora | Training | 0.16 | 0.22 |
| | Testing | 0.12 | 0.16 |

## F   LIMITATIONS OF OUR WORK

Our SPD network GyroSpd++ relies on different Riemannian metrics across the layers, i.e., the convolutional layer is based on Affine-Invariant metrics while the MLR layer is based on Log-Euclidean metrics. Although we have provided the experimental results demonstrating that GyroSpd++ achieves good performance on all the datasets compared to state-of-the-art methods, it is not clear if our design is optimal for the human action recognition task. When it comes to building a deep SPD architecture, it is useful to provide insights into Riemannian metrics one should use for each network block in order to obtain good performance on a target task.

In our Grassmann network Gr-GCN++, the feature transformation and bias and nonlinearity operations are performed on Grassmann manifolds, while the aggregation operation is performed in tangent spaces. Previous works (Dai et al., 2021; Chen et al., 2022) on HNNs have shown that this hybrid method limits the modeling ability of networks. Therefore, it is desirable to develop GCNs where all the operations are formalized on Grassmann manifolds.

Table 14: Results (mean accuracy $\pm$ standard deviation) of Gr-GCN++ and some state-of-the-art methods on the three datasets. The best and second best results in terms of mean accuracy are highlighted in red and blue, respectively.

| Method | Airport | Pubmed | Cora |
|---|---|---|---|
| GCN (Kipf & Welling, 2017) | $82.2 \pm 0.6$ | $77.8 \pm 0.8$ | $80.2 \pm 2.3$ |
| GAT (Veličković et al., 2018) | $\textbf{\textcolor{blue}{92.9}} \pm 0.8$ | $77.6 \pm 0.8$ | $80.3 \pm 0.6$ |
| HGNN (Liu et al., 2019) | $84.5 \pm 0.7$ | $76.6 \pm 1.4$ | $79.5 \pm 0.9$ |
| HGCN (Chami et al., 2019) | $85.3 \pm 0.6$ | $76.4 \pm 0.8$ | $78.7 \pm 0.9$ |
| LGCN (Zhang et al., 2021) | $88.2 \pm 0.2$ | $77.3 \pm 1.4$ | $80.6 \pm 0.9$ |
| HGAT (Zhang et al., 2022) | $87.5 \pm 0.9$ | $78.0 \pm 0.5$ | $80.9 \pm 0.7$ |
| SPD-GCN (Zhao et al., 2023) | $82.6 \pm 1.5$ | $\textbf{\textcolor{blue}{78.7}} \pm 0.5$ | $\textbf{\textcolor{red}{82.3}} \pm 0.5$ |
| HamGNN (Kang et al., 2023) | $\textbf{\textcolor{red}{95.9}} \pm 0.1$ | $78.3 \pm 0.6$ | $80.1 \pm 1.6$ |
| Gr-GCN++ (Ours) | $82.8 \pm 0.7$ | $\textbf{\textcolor{red}{80.3}} \pm 0.5$ | $\textbf{\textcolor{blue}{81.6}} \pm 0.4$ |

## G   SOME RELATED DEFINITIONS

### G.1   GYROGROUPS AND GYROVECTOR SPACES

Gyrovector spaces form the setting for hyperbolic geometry in the same way that vector spaces form the setting for Euclidean geometry (Ungar, 2002; 2005; 2014). We recap the definitions of gyrogroups and gyrocommutative gyrogroups proposed in Ungar (2002; 2005; 2014). For greater mathematical detail and in-depth discussion, we refer the interested reader to these papers.

**Definition G.1** (**Gyrogroups (Ungar, 2014)**). *A pair $(G, \oplus)$ is a groupoid in the sense that it is a nonempty set, $G$, with a binary operation, $\oplus$. A groupoid $(G, \oplus)$ is a gyrogroup if its binary operation satisfies the following axioms for $a, b, c \in G$:*

*(G1) There is at least one element $e \in G$ called a left identity such that $e \oplus a = a$.*

*(G2) There is an element $\ominus a \in G$ called a left inverse of $a$ such that $\ominus a \oplus a = e$.*

*(G3) There is an automorphism $\mathrm{gyr}[a, b] : G \to G$ for each $a, b \in G$ such that*

$$a \oplus (b \oplus c) = (a \oplus b) \oplus \mathrm{gyr}[a, b]c \quad \text{(Left Gyroassociative Law)}.$$

*The automorphism $\mathrm{gyr}[a, b]$ is called the gyroautomorphism, or the gyration of $G$ generated by $a, b$.*

*(G4) $\mathrm{gyr}[a, b] = \mathrm{gyr}[a \oplus b, b]$ (Left Reduction Property).*

**Definition G.2** (**Gyrocommutative Gyrogroups (Ungar, 2014)**). *A gyrogroup $(G, \oplus)$ is gyrocommutative if it satisfies*

$$a \oplus b = \mathrm{gyr}[a, b](b \oplus a) \quad \text{(Gyrocommutative Law)}.$$

The following definition of gyrovector spaces is slightly different from Definition 3.2 in Ungar (2014).

**Definition G.3** (**Gyrovector Spaces**). *A gyrocommutative gyrogroup $(G, \oplus)$ equipped with a scalar multiplication*

$$(t, x) \to t \odot x : \mathbb{R} \times G \to G$$

*is called a gyrovector space if it satisfies the following axioms for $s, t \in \mathbb{R}$ and $a, b, c \in G$:*

*(V1) $1 \odot a = a, 0 \odot a = t \odot e = e$, and $(-1) \odot a = \ominus a$.*

*(V2) $(s + t) \odot a = s \odot a \oplus t \odot a$.*

*(V3) $(st) \odot a = s \odot (t \odot a)$.*

*(V4) $\mathrm{gyr}[a, b](t \odot c) = t \odot \mathrm{gyr}[a, b]c$.*

*(V5) $\mathrm{gyr}[s \odot a, t \odot a] = \mathrm{Id}$, where $\mathrm{Id}$ is the identity map.*

### G.2 AI GYROVECTOR SPACES

For $\mathbf{P}, \mathbf{Q} \in \mathrm{Sym}_n^+$, the binary operation (Nguyen, 2022a) is given as

$$\mathbf{P} \oplus_{ai} \mathbf{Q} = \mathbf{P}^{\frac{1}{2}} \mathbf{Q} \mathbf{P}^{\frac{1}{2}}.$$

The inverse operation (Nguyen, 2022a) is given by

$$\ominus_{ai}\mathbf{P} = \mathbf{P}^{-1}.$$

### G.3 LE GYROVECTOR SPACES

For $\mathbf{P}, \mathbf{Q} \in \mathrm{Sym}_n^+$, the binary operation (Nguyen, 2022a) is given as

$$\mathbf{P} \oplus_{le} \mathbf{Q} = \exp(\log(\mathbf{P}) + \log(\mathbf{Q})).$$

The inverse operation (Nguyen, 2022a) is given as

$$\ominus_{le}\mathbf{P} = \mathbf{P}^{-1}.$$

### G.4 LC GYROVECTOR SPACES

For $\mathbf{P}, \mathbf{Q} \in \mathrm{Sym}_n^+$, the binary operation (Nguyen, 2022a) is given as

$$\mathbf{P} \oplus_{lc} \mathbf{Q} = \big(\lfloor \mathscr{L}(\mathbf{P}) \rfloor + \lfloor \mathscr{L}(\mathbf{Q}) \rfloor + \mathbb{D}(\mathscr{L}(\mathbf{P}))\mathbb{D}(\mathscr{L}(\mathbf{Q}))\big).\big(\lfloor \mathscr{L}(\mathbf{P}) \rfloor + \lfloor \mathscr{L}(\mathbf{Q}) \rfloor + \mathbb{D}(\mathscr{L}(\mathbf{P}))\mathbb{D}(\mathscr{L}(\mathbf{Q}))\big)^T,$$

where $\lfloor \mathbf{Y} \rfloor$ is a matrix of the same size as matrix $\mathbf{Y} \in \mathrm{M}_{n,n}$ whose $(i,j)$ element is $\mathbf{Y}_{(i,j)}$ if $i > j$ and is zero otherwise, $\mathbb{D}(\mathbf{Y})$ is a diagonal matrix of the same size as matrix $\mathbf{Y}$ whose $(i,i)$ element is $\mathbf{Y}_{(i,i)}$, and $\mathscr{L}(\mathbf{P})$ denotes the Cholesky factor of $\mathbf{P}$, i.e., $\mathscr{L}(\mathbf{P})$ is a lower triangular matrix with positive diagonal entries such that $\mathbf{P} = \mathscr{L}(\mathbf{P})\mathscr{L}(\mathbf{P})^T$.

The inverse operation (Nguyen, 2022a) is given by

$$\ominus_{lc}\mathbf{P} = \big(-\lfloor \mathscr{L}(\mathbf{P}) \rfloor + \mathbb{D}(\mathscr{L}(\mathbf{P}))^{-1}\big)\big(-\lfloor \mathscr{L}(\mathbf{P}) \rfloor + \mathbb{D}(\mathscr{L}(\mathbf{P}))^{-1}\big)^T.$$

### G.5 GRASSMANN MANIFOLDS IN THE PROJECTOR PERSPECTIVE

For $\mathbf{P}, \mathbf{Q} \in \mathrm{Gr}_{n,p}$, the binary operation (Nguyen, 2022b) is given as

$$\mathbf{P} \oplus_{gr} \mathbf{Q} = \exp([\mathrm{Log}_{\mathbf{I}_{n,p}}^{gr}(\mathbf{P}), \mathbf{I}_{n,p}])\mathbf{Q} \exp(-[\mathrm{Log}_{\mathbf{I}_{n,p}}^{gr}(\mathbf{P}), \mathbf{I}_{n,p}]),$$

where $[.,.]$ denotes the matrix commutator.

The inverse operation (Nguyen, 2022b) is defined as

$$\ominus_{gr}\mathbf{P} = \mathrm{Exp}_{\mathbf{I}_{n,p}}^{gr}(-\mathrm{Log}_{\mathbf{I}_{n,p}}^{gr}(\mathbf{P})).$$

### G.6 GRASSMANN MANIFOLDS IN THE ONB PERSPECTIVE

For $\mathbf{U}, \mathbf{V} \in \widetilde{\mathrm{Gr}}_{n,p}$, the binary operation (Nguyen & Yang, 2023) is defined as

$$\mathbf{U} \widetilde{\oplus}_{gr} \mathbf{V} = \exp([\mathrm{Log}_{\mathbf{I}_{n,p}}^{gr}(\mathbf{U}\mathbf{U}^T), \mathbf{I}_{n,p}])\mathbf{V}.$$

The inverse operation can be defined using the approach in Nguyen & Yang (2023) (see Section 2.3.1), i.e.,

$$\widetilde{\ominus}_{gr}\mathbf{U} = \tau^{-1}\big(\ominus_{gr}(\mathbf{U}\mathbf{U}^T)\big),$$

where the mapping $\tau$ is defined in Proposition 3.12, i.e.,

$$\tau : \widetilde{\mathrm{Gr}}_{n,p} \to \mathrm{Gr}_{n,p}, \ \mathbf{U} \mapsto \mathbf{U}\mathbf{U}^T.$$

### G.7 THE SPD AND GRASSMANN INNER PRODUCTS

**Definition G.4 (The SPD Inner Product).** *Let* $\mathbf{P}, \mathbf{Q} \in \mathrm{Sym}_n^{+,g}$. *Then the SPD inner product of* $\mathbf{P}$ *and* $\mathbf{Q}$ *is defined as*

$$\langle \mathbf{P}, \mathbf{Q} \rangle^g = \langle \mathrm{Log}_{\mathbf{I}_n}^g(\mathbf{P}), \mathrm{Log}_{\mathbf{I}_n}^g(\mathbf{Q}) \rangle_{\mathbf{I}_n}^g.$$

**Definition G.5 (The Grassmann Inner Product).** *Let* $\mathbf{P}, \mathbf{Q} \in \mathrm{Gr}_{n,p}$. *Then the Grassmann inner product of* $\mathbf{P}$ *and* $\mathbf{Q}$ *is defined as*

$$\langle \mathbf{P}, \mathbf{Q} \rangle^{gr} = \langle \mathrm{Log}_{\mathbf{I}_{n,p}}^{gr}(\mathbf{P}), \mathrm{Log}_{\mathbf{I}_{n,p}}^{gr}(\mathbf{Q}) \rangle_{\mathbf{I}_{n,p}},$$

*where* $\langle ., . \rangle_{\mathbf{I}_{n,p}}$ *denotes the inner product at* $\mathbf{I}_{n,p}$ *given by the canonical metric of* $\mathrm{Gr}_{n,p}$.

### G.8 THE GYROCOSINE FUNCTION AND GYROANGLES IN STRUCTURE SPACES

**Definition G.6 (The Gyrocosine Function and Gyroangles).** *Let* $\mathbf{P}, \mathbf{Q}$, *and* $\mathbf{R}$ *be three distinct gyropoints in structure space* $\widetilde{\mathrm{Gr}}_{n,p} \times \mathrm{Sym}_p^+$. *The gyrocosine of the measure of the gyroangle* $\alpha$, $0 \leq \alpha \leq \pi$, *between* $\ominus_{psd,g} \mathbf{P} \oplus_{psd,g} \mathbf{Q}$ *and* $\ominus_{psd,g} \mathbf{P} \oplus_{psd,g} \mathbf{R}$ *is given by the equation*

$$\cos \alpha = \frac{\langle \ominus_{psd,g} \mathbf{P} \oplus_{psd,g} \mathbf{Q}, \ominus_{psd,g} \mathbf{P} \oplus_{psd,g} \mathbf{R} \rangle^{psd,g}}{\| \ominus_{psd,g} \mathbf{P} \oplus_{psd,g} \mathbf{Q} \|^{psd,g} . \| \ominus_{psd,g} \mathbf{P} \oplus_{psd,g} \mathbf{R} \|^{psd,g}},$$

*where* $\|.\|^{psd,g}$ *is the norm induced by the inner product in structure spaces. The gyroangle* $\alpha$ *is denoted by* $\alpha = \angle \mathbf{QPR}$.

### G.9 THE GYRODISTANCE FUNCTION IN STRUCTURE SPACES

**Definition G.7 (The Gyrodistance Function in Structure Spaces).** *Let* $\mathbf{P}, \mathbf{Q} \in \widetilde{\mathrm{Gr}}_{n,p} \times \mathrm{Sym}_p^+$. *Then the gyrodistance function in structure spaces* $\widetilde{\mathrm{Gr}}_{n,p} \times \mathrm{Sym}_p^+$ *is defined as*

$$d(\mathbf{P}, \mathbf{Q}) = \| \ominus_{psd,g} \mathbf{P} \oplus_{psd,g} \mathbf{Q} \|^{psd,g}.$$

### G.10 THE PSEUDO-GYRODISTANCE FUNCTION IN STRUCTURE SPACES

**Definition G.8 (The Pseudo-gyrodistance Function in Structure Spaces).** *Let* $\mathcal{H}_{\mathbf{W},\mathbf{P}}^{psd,g}$ *be a hyper-gyroplane in structure space* $\widetilde{\mathrm{Gr}}_{n,p} \times \mathrm{Sym}_p^+$, *and* $\mathbf{X} \in \widetilde{\mathrm{Gr}}_{n,p} \times \mathrm{Sym}_p^+$. *Then the pseudo-gyrodistance from* $\mathbf{X}$ *to* $\mathcal{H}_{\mathbf{W},\mathbf{P}}^{psd,g}$ *is defined as*

$$\bar{d}(\mathbf{X}, \mathcal{H}_{\mathbf{W},\mathbf{P}}^{psd,g}) = \sin(\angle \mathbf{X} \mathbf{P} \overline{\mathbf{Q}}) d(\mathbf{X}, \mathbf{P}),$$

*where* $\overline{\mathbf{Q}}$ *is given by*

$$\overline{\mathbf{Q}} = \underset{\mathbf{Q} \in \mathcal{H}_{\mathbf{W},\mathbf{P}}^{psd,g} \setminus \{\mathbf{P}\}}{\arg \max} \cos(\angle \mathbf{Q} \mathbf{P} \mathbf{X}).$$

*By convention,* $\sin(\angle \mathbf{X} \mathbf{P} \mathbf{Q}) = 0$ *for any* $\mathbf{X}, \mathbf{Q} \in \mathcal{H}_{\mathbf{W},\mathbf{P}}^{psd,g}$.

## H COMPUTATION OF CANONICAL REPRESENTATIONS

Let $\mathrm{V}_{n,p}$ be the space of $n \times p$ matrices with orthonormal columns. For any $\mathbf{P} \in \mathrm{S}_{n,p}^+$, let $\mathbf{U}_P \in \widetilde{\mathrm{Gr}}_{n,p}$, $\mathbf{S}_P \in \mathrm{Sym}_p^+$ such that $\mathbf{P} = \mathbf{U}_P \mathbf{S}_P \mathbf{U}_P^T$. Denote by $\mathbf{W}$ the common subspace used for computing a canonical representation of $\mathbf{P}$. We first compute two bases of $\mathrm{span}(\mathbf{U}_P)$ and $\mathrm{span}(\mathbf{W})$, denoted respectively by $\overline{\mathbf{U}}$ and $\overline{\mathbf{W}}$, such that

$$d_{\mathrm{V}_{n,p}}(\overline{\mathbf{U}}, \overline{\mathbf{W}}) = d_{\widetilde{\mathrm{Gr}}_{n,p}}(\mathrm{span}(\mathbf{U}_P), \mathrm{span}(\mathbf{W})),$$

where $d_{\mathrm{V}_{n,p}}(.,.)$ and $d_{\widetilde{\mathrm{Gr}}_{n,p}}(.,.)$ are the distances between two points in $\mathrm{V}_{n,p}$ and $\widetilde{\mathrm{Gr}}_{n,p}$, respectively. These two bases can be computed as $\overline{\mathbf{U}} = \mathbf{U}_P \mathbf{Y}$, $\overline{\mathbf{W}} = \mathbf{W}\mathbf{V}$, where $\mathbf{Y}$ and $\mathbf{V}$ are obtained from a SVD of $(\mathbf{U}_P)^T \mathbf{W}$, i.e.,

$$(\mathbf{U}_P)^T \mathbf{W} = \mathbf{Y}(\cos \mathbf{\Sigma})\mathbf{V}^T.$$

The SPD matrix $\overline{\mathbf{S}}_P$ in the canonical representation of $\mathbf{P}$ is then computed as

$$\overline{\mathbf{S}}_P = \mathbf{V}\overline{\mathbf{U}}^T \mathbf{P}\overline{\mathbf{U}}\mathbf{V}^T.$$

## I  PROOF OF PROPOSITION 3.2

*Proof.* We first recall the definition of the binary operation $\oplus_g$ in Nguyen (2022b).

**Definition I.1 (The Binary Operation (Nguyen, 2022b)).** *Let* $\mathbf{P}, \mathbf{Q} \in \mathrm{Sym}_n^+$. *Then the binary operation* $\oplus_g$ *is defined as*

$$\mathbf{P} \oplus_g \mathbf{Q} = \mathrm{Exp}_{\mathbf{P}}^g(\mathcal{T}_{\mathbf{I}_n \to \mathbf{P}}^g(\mathrm{Log}_{\mathbf{I}_n}^g(\mathbf{Q}))).$$

We have

$$
\begin{aligned}
\langle \mathrm{Log}_{\mathbf{P}}^g(\mathbf{Q}), \mathbf{W} \rangle_{\mathbf{P}}^g &\overset{(1)}{=} \langle \mathcal{T}_{\mathbf{P} \to \mathbf{I}_n}^g\big(\mathrm{Log}_{\mathbf{P}}^g(\mathbf{Q})\big), \mathcal{T}_{\mathbf{P} \to \mathbf{I}_n}^g(\mathbf{W}) \rangle_{\mathbf{I}_n}^g \\
&\overset{(2)}{=} \langle \mathrm{Exp}_{\mathbf{I}_n}^g\left(\mathcal{T}_{\mathbf{P} \to \mathbf{I}_n}^g\big(\mathrm{Log}_{\mathbf{P}}^g(\mathbf{Q})\big)\right), \mathrm{Exp}_{\mathbf{I}_n}^g\left(\mathcal{T}_{\mathbf{P} \to \mathbf{I}_n}^g(\mathbf{W})\right) \rangle^g,
\end{aligned}
\tag{5}
$$

where (1) follows from the invariance of the inner product under parallel transport, and (2) follows from Definition G.4.

Let $\mathbf{R} = \mathrm{Exp}_{\mathbf{I}_n}^g\left(\mathcal{T}_{\mathbf{P} \to \mathbf{I}_n}^g\big(\mathrm{Log}_{\mathbf{P}}^g(\mathbf{Q})\big)\right)$. Then

$$\mathrm{Log}_{\mathbf{I_n}}^g(\mathbf{R}) = \mathcal{T}_{\mathbf{P} \to \mathbf{I}_n}^g\big(\mathrm{Log}_{\mathbf{P}}^g(\mathbf{Q})\big),$$

which results in

$$\mathcal{T}_{\mathbf{I}_n \to \mathbf{P}}^g\big(\mathrm{Log}_{\mathbf{I_n}}^g(\mathbf{R})\big) = \mathrm{Log}_{\mathbf{P}}^g(\mathbf{Q}).$$

Hence

$$\mathrm{Exp}_{\mathbf{P}}^g\left(\mathcal{T}_{\mathbf{I}_n \to \mathbf{P}}^g\big(\mathrm{Log}_{\mathbf{I_n}}^g(\mathbf{R})\big)\right) = \mathbf{Q}.$$

By the Left Cancellation Law,

$$\mathbf{Q} = \mathbf{P} \oplus_g (\ominus_g \mathbf{P} \oplus_g \mathbf{Q}).$$

Therefore

$$
\begin{aligned}
\mathrm{Exp}_{\mathbf{P}}^g\left(\mathcal{T}_{\mathbf{I}_n \to \mathbf{P}}^g\big(\mathrm{Log}_{\mathbf{I_n}}^g(\mathbf{R})\big)\right) &= \mathbf{P} \oplus_g (\ominus_g \mathbf{P} \oplus_g \mathbf{Q}) \\
&= \mathrm{Exp}_{\mathbf{P}}^g\left(\mathcal{T}_{\mathbf{I}_n \to \mathbf{P}}^g\big(\mathrm{Log}_{\mathbf{I_n}}^g(\ominus_g \mathbf{P} \oplus_g \mathbf{Q})\big)\right),
\end{aligned}
$$

where the last equality follows from Definition I.1.

We thus have

$$
\begin{aligned}
\ominus_g \mathbf{P} \oplus_g \mathbf{Q} &= \mathbf{R} \\
&= \mathrm{Exp}_{\mathbf{I}_n}^g\left(\mathcal{T}_{\mathbf{P} \to \mathbf{I}_n}^g\big(\mathrm{Log}_{\mathbf{P}}^g(\mathbf{Q})\big)\right).
\end{aligned}
\tag{6}
$$

Combining Eqs. (5) and (6), we get

$$\langle \mathrm{Log}_{\mathbf{P}}^g(\mathbf{Q}), \mathbf{W} \rangle_{\mathbf{P}}^g = \langle \ominus_g \mathbf{P} \oplus_g \mathbf{Q}, \mathrm{Exp}_{\mathbf{I}_n}^g\left(\mathcal{T}_{\mathbf{P} \to \mathbf{I}_n}^g(\mathbf{W})\right) \rangle^g,$$

which concludes the proof of Proposition 3.2.

$\square$

## J  PROOF OF PROPOSITION 3.4

*Proof.* The first part of Proposition 3.4 can be easily verified using the definition of the SPD inner product (see Definition G.4) and that of Affine-Invariant metrics (Pennec et al., 2020) (see Chapter 3).

To prove the second part of Proposition 3.4, we will use the notion of SPD pseudo-gyrodistance (Nguyen & Yang, 2023) in our interpretation of FC layers on SPD manifolds, i.e., the signed distance is replaced with the signed SPD pseudo-gyrodistance in the interpretation given in Section 3.2.1. First, we need the following result from Nguyen & Yang (2023).

**Theorem J.1 (The SPD Pseudo-gyrodistance from an SPD Matrix to an SPD Hypergyroplane in an AI Gyrovector Space (Nguyen & Yang, 2023)).** *Let $\mathcal{H}_{\mathbf{W},\mathbf{P}}$ be an SPD hypergyroplane in a gyrovector space $(\mathrm{Sym}_n^+, \oplus_{ai}, \otimes_{ai})$, and $\mathbf{X} \in \mathrm{Sym}_n^+$. Then the SPD pseudo-gyrodistance from $\mathbf{X}$ to $\mathcal{H}_{\mathbf{W},\mathbf{P}}$ is given by*

$$\bar{d}(\mathbf{X}, \mathcal{H}_{\mathbf{W},\mathbf{P}}) = \frac{|\langle \log(\mathbf{P}^{-\frac{1}{2}}\mathbf{X}\mathbf{P}^{-\frac{1}{2}}), \mathbf{P}^{-\frac{1}{2}}\mathbf{W}\mathbf{P}^{-\frac{1}{2}}\rangle_F|}{\|\mathbf{P}^{-\frac{1}{2}}\mathbf{W}\mathbf{P}^{-\frac{1}{2}}\|_F}.$$

By Theorem J.1, the signed SPD pseudo-gyrodistance from $\mathbf{Y}$ to an SPD hypergyroplane that contains the origin and is orthogonal to the $E_{(i,j)}^{ai}$ axis is given by

$$\bar{d}(\mathbf{Y}, \mathcal{H}_{\mathrm{Log}_{\mathbf{I}_m}^{ai}(E_{(i,j)}^{ai}), \mathbf{I}_m}) = \frac{\langle \log(\mathbf{Y}), \mathrm{Log}_{\mathbf{I}_m}^{ai}(E_{(i,j)}^{ai})\rangle_F}{\|\mathrm{Log}_{\mathbf{I}_m}^{ai}(E_{(i,j)}^{ai})\|_F}.$$

According to our interpretation of FC layers,

$$v_{(i,j)}(\mathbf{X}) = \frac{\langle \log(\mathbf{Y}), \mathrm{Log}_{\mathbf{I}_m}^{ai}(E_{(i,j)}^{ai})\rangle_F}{\|\mathrm{Log}_{\mathbf{I}_m}^{ai}(E_{(i,j)}^{ai})\|_F}.$$

We consider two cases:

*Case 1: $i < j$.*

$$\begin{aligned}
v_{(i,j)}(\mathbf{X}) &= \frac{\langle \log(\mathbf{Y}), \frac{1}{\sqrt{2}}(\mathbf{e}_i\mathbf{e}_j^T + \mathbf{e}_j\mathbf{e}_i^T)\rangle_F}{\|\frac{1}{\sqrt{2}}(\mathbf{e}_i\mathbf{e}_j^T + \mathbf{e}_j\mathbf{e}_i^T)\|_F} \\
&= \langle \log(\mathbf{Y}), \frac{1}{\sqrt{2}}(\mathbf{e}_i\mathbf{e}_j^T + \mathbf{e}_j\mathbf{e}_i^T)\rangle_F \\
&= \frac{1}{\sqrt{2}}\big(\log(\mathbf{Y})_{(i,j)} + \log(\mathbf{Y})_{(j,i)}\big) \\
&= \sqrt{2}\log(\mathbf{Y})_{(i,j)}.
\end{aligned}$$

We thus deduce that

$$\log(\mathbf{Y})_{(i,j)} = \frac{1}{\sqrt{2}}v_{(i,j)}(\mathbf{X}).$$

*Case 2: $i = j$.*

$$\begin{aligned}
v_{(i,i)}(\mathbf{X}) &= \frac{\langle \log(\mathbf{Y}), \mathbf{e}_i\mathbf{e}_i^T - \frac{1}{m}\big(1 - \frac{1}{\sqrt{1+m\beta}}\big)\mathbf{I}_m\rangle_F}{\|\mathbf{e}_i\mathbf{e}_i^T - \frac{1}{m}\big(1 - \frac{1}{\sqrt{1+m\beta}}\big)\mathbf{I}_m\|_F} \\
&= \langle \log(\mathbf{Y}), \mathbf{e}_i\mathbf{e}_i^T - \frac{1}{m}\big(1 - \frac{1}{\sqrt{1+m\beta}}\big)\mathbf{I}_m\rangle_F.
\end{aligned}$$

This leads to

$$v_{(i,i)}(\mathbf{X}) = \log(\mathbf{Y})_{(i,i)} - \frac{1}{m}\Big(1 - \frac{1}{\sqrt{1+m\beta}}\Big)\sum_{j=1}^{m}\log(\mathbf{Y})_{(j,j)}, \tag{7}$$

for $i = 1, \ldots, m$. By summing up $v_{(i,i)}(\mathbf{X}), i = 1, \ldots, m$, we get

$$\sum_{i=1}^{m} v_{(i,i)}(\mathbf{X}) = \frac{1}{\sqrt{1+m\beta}}\sum_{i=1}^{m}\log(\mathbf{Y})_{(i,i)},$$

or equivalently,

$$\sum_{i=1}^{m}\log(\mathbf{Y})_{(i,i)} = \sqrt{1+m\beta}\Big(\sum_{i=1}^{m} v_{(i,i)}(\mathbf{X})\Big). \tag{8}$$

Replacing the term $\sum_{j=1}^{m}\log(\mathbf{Y})_{(j,j)}$ in Eq. (7) with the expression on the right-hand side of Eq. (8) results in

$$\log(\mathbf{Y})_{(i,i)} = v_{(i,i)}(\mathbf{X}) + \frac{1}{m}(\sqrt{1+m\beta}-1)\sum_{j=1}^{m} v_{(j,j)}(\mathbf{X}).$$

Note that $\mathbf{Y} = \exp([\log(\mathbf{Y})_{(i,j)}]_{i,j=1}^{m})$. This concludes the proof of Proposition 3.4.

$\square$

## K    PROOF OF PROPOSITION 3.5

*Proof.* This proposition is a direct consequence of Proposition 3.4 for $\beta = 0$. $\square$

## L    PROOF OF PROPOSITION 3.6

*Proof.* The first part of Proposition 3.6 can be easily verified using the definition of the SPD inner product (see Definition G.4) and that of Log-Cholesky metrics (Lin, 2019).

To prove the second part of Proposition 3.6, we first recall the following result from Nguyen & Yang (2023).

**Theorem L.1 (The SPD Gyrodistance from an SPD Matrix to an SPD Hypergyroplane in a LC Gyrovector Space (Nguyen & Yang, 2023)).** *Let $\mathcal{H}_{\mathbf{W},\mathbf{P}}$ be an SPD hypergyroplane in a gyrovector space $(\mathrm{Sym}_n^+, \oplus_{lc}, \otimes_{lc})$, and $\mathbf{X} \in \mathrm{Sym}_n^+$. Then the SPD pseudo-gyrodistance from $\mathbf{X}$ to $\mathcal{H}_{\mathbf{W},\mathbf{P}}$ is equal to the SPD gyrodistance from $\mathbf{X}$ to $\mathcal{H}_{\mathbf{W},\mathbf{P}}$ and is given by*

$$d(\mathbf{X}, \mathcal{H}_{\mathbf{W},\mathbf{P}}) = \frac{|\langle \mathbf{A}, \mathbf{B}\rangle_F|}{\|\mathbf{B}\|_F},$$

*where*

$$\mathbf{A} = -\lfloor\varphi(\mathbf{P})\rfloor + \lfloor\varphi(\mathbf{X})\rfloor + \log(\mathbb{D}(\varphi(\mathbf{P}))^{-1}\mathbb{D}(\varphi(\mathbf{X}))),$$

$$\mathbf{B} = \lfloor\widetilde{\mathbf{W}}\rfloor + \mathbb{D}(\varphi(\mathbf{P}))^{-1}\mathbb{D}(\widetilde{\mathbf{W}}),$$

$$\widetilde{\mathbf{W}} = \varphi(\mathbf{P})\Big(\varphi(\mathbf{P})^{-1}\mathbf{W}(\varphi(\mathbf{P})^{-1})^T\Big)_{\frac{1}{2}},$$

*where $\lfloor\mathbf{Y}\rfloor$ and $\mathbb{D}(\mathbf{Y}), \mathbf{Y} \in \mathrm{M}_{n,n}$ are defined in Section G.4, and $\varphi(\mathbf{P}) = \mathscr{L}(\mathbf{P})$.*

By Theorem L.1, the signed SPD pseudo-gyrodistance from $\mathbf{Y}$ to an SPD hypergyroplane that contains the origin and is orthogonal to the $E_{(i,j)}^{lc}$ axis is given by

$$d(\mathbf{Y}, \mathcal{H}_{\mathrm{Log}_{\mathbf{I}_m}^{lc}(E_{(i,j)}^{lc}), \mathbf{I}_m}) = \frac{\langle \lfloor\varphi(\mathbf{Y})\rfloor + \log(\mathbb{D}(\varphi(\mathbf{Y}))), \big(\mathrm{Log}_{\mathbf{I}_m}^{lc}(E_{(i,j)}^{lc})\big)_{\frac{1}{2}}\rangle_F}{\big\|\big(\mathrm{Log}_{\mathbf{I}_m}^{lc}(E_{(i,j)}^{lc})\big)_{\frac{1}{2}}\big\|_F}.$$

According to our interpretation of FC layers,

$$v_{(i,j)}(\mathbf{X}) = \frac{\langle \lfloor \varphi(\mathbf{Y}) \rfloor + \log(\mathbb{D}(\varphi(\mathbf{Y}))), \left( \mathrm{Log}_{\mathbf{I}_m}^{lc}(E_{(i,j)}^{lc}) \right)_{\frac{1}{2}} \rangle_F}{\| \left( \mathrm{Log}_{\mathbf{I}_m}^{lc}(E_{(i,j)}^{lc}) \right)_{\frac{1}{2}} \|_F}.$$

We consider two cases:

*Case 1: $i < j$.*

$$
\begin{aligned}
v_{(i,j)}(\mathbf{X}) &= \frac{\langle \lfloor \varphi(\mathbf{Y}) \rfloor + \log(\mathbb{D}(\varphi(\mathbf{Y}))), \mathbf{e}_j \mathbf{e}_i^T \rangle_F}{\| \mathbf{e}_j \mathbf{e}_i^T \|_F} \\
&= \langle \lfloor \varphi(\mathbf{Y}) \rfloor + \log(\mathbb{D}(\varphi(\mathbf{Y}))), \mathbf{e}_j \mathbf{e}_i^T \rangle_F \\
&= \varphi(\mathbf{Y})_{(j,i)}.
\end{aligned}
$$

We thus have

$$\varphi(\mathbf{Y})_{(j,i)} = v_{(i,j)}(\mathbf{X}).$$

*Case 2: $i = j$.*

$$
\begin{aligned}
v_{(i,j)}(\mathbf{X}) &= \frac{\langle \lfloor \varphi(\mathbf{Y}) \rfloor + \log(\mathbb{D}(\varphi(\mathbf{Y}))), \mathbf{e}_i \mathbf{e}_i^T \rangle_F}{\| \mathbf{e}_i \mathbf{e}_i^T \|_F} \\
&= \langle \lfloor \varphi(\mathbf{Y}) \rfloor + \log(\mathbb{D}(\varphi(\mathbf{Y}))), \mathbf{e}_i \mathbf{e}_i^T \rangle_F \\
&= \log(\varphi(\mathbf{Y})_{(i,i)}).
\end{aligned}
$$

Hence

$$\varphi(\mathbf{Y})_{(i,i)} = \exp(v_{(i,i)}(\mathbf{X})).$$

Setting $\varphi(\mathbf{Y}) = [y_{(i,j)}]_{i,j=1}^m$, then $y_{(i,j)}$ are given by

$$
y_{(j,i)} = \begin{cases} \exp(v_{(i,j)}(\mathbf{X})), & \text{if } i = j \\ v_{(i,j)}(\mathbf{X}), & \text{if } i < j \\ 0, & \text{if } i > j \end{cases}
$$

Since $\varphi(\mathbf{Y})$ is the Cholesky factor of $\mathbf{Y}$, we have

$$\mathbf{Y} = \varphi(\mathbf{Y})\varphi(\mathbf{Y})^T,$$

which concludes the proof of Proposition 3.6.

$\square$

## M  PROOF OF THEOREM 3.11

*Proof.* Let $\mathcal{H}_{\mathbf{W},\mathbf{P}}^{psd,g}$ be a hypergyroplane in structure space $\widetilde{\mathrm{Gr}}_{n,p} \times \mathrm{Sym}_p^+$ and $\mathbf{X} \in \widetilde{\mathrm{Gr}}_{n,p} \times \mathrm{Sym}_p^+$. By the definition of the pseudo-gyrodistance function,

$$\bar{d}(\mathbf{X}, \mathcal{H}_{\mathbf{W},\mathbf{P}}^{psd,g}) = \sin(\angle \mathbf{X}\mathbf{P}\overline{\mathbf{Q}}) d(\mathbf{X}, \mathbf{P}),$$

where $\overline{\mathbf{Q}}$ is given by

$$
\begin{aligned}
\overline{\mathbf{Q}} &= \mathop{\arg\max}_{\mathbf{Q} \in \mathcal{H}_{\mathbf{W},\mathbf{P}}^{psd,g} \setminus \{\mathbf{P}\}} \cos(\angle \mathbf{Q}\mathbf{P}\mathbf{X}) \\
&= \mathop{\arg\max}_{\mathbf{Q} \in \mathcal{H}_{\mathbf{W},\mathbf{P}}^{psd,g} \setminus \{\mathbf{P}\}} \frac{\langle \ominus_{psd,g}\mathbf{P} \oplus_{psd,g} \mathbf{Q}, \ominus_{psd,g}\mathbf{P} \oplus_{psd,g} \mathbf{X} \rangle^{psd,g}}{\| \ominus_{psd,g} \mathbf{P} \oplus_{psd,g} \mathbf{Q} \|^{psd,g}. \| \ominus_{psd,g} \mathbf{P} \oplus_{psd,g} \mathbf{X} \|^{psd,g}}.
\end{aligned}
$$

By the definitions of the binary and inverse operations in structure spaces,

$$\ominus_{psd,g}\mathbf{P}\oplus_{psd,g}\mathbf{X} = (\widetilde{\ominus}_{gr}\mathbf{U}_P\widetilde{\oplus}_{gr}\mathbf{U}_X, \ominus_g\mathbf{S}_P\oplus_g\mathbf{S}_X),$$

$$\ominus_{psd,g}\mathbf{P}\oplus_{psd,g}\mathbf{Q} = (\widetilde{\ominus}_{gr}\mathbf{U}_P\widetilde{\oplus}_{gr}\mathbf{U}_Q, \ominus_g\mathbf{S}_P\oplus_g\mathbf{S}_Q).$$

Hence

$$\langle\ominus_{psd,g}\mathbf{P}\oplus_{psd,g}\mathbf{X}, \ominus_{psd,g}\mathbf{P}\oplus_{psd,g}\mathbf{Q}\rangle^{psd,g} = \lambda\langle(\widetilde{\ominus}_{gr}\mathbf{U}_P\widetilde{\oplus}_{gr}\mathbf{U}_X)(\widetilde{\ominus}_{gr}\mathbf{U}_P\widetilde{\oplus}_{gr}\mathbf{U}_X)^T,$$
$$(\widetilde{\ominus}_{gr}\mathbf{U}_P\widetilde{\oplus}_{gr}\mathbf{U}_Q)(\widetilde{\ominus}_{gr}\mathbf{U}_P\widetilde{\oplus}_{gr}\mathbf{U}_Q)^T\rangle^{gr}$$
$$+ \langle\ominus_g\mathbf{S}_P\oplus_g\mathbf{S}_X, \ominus_g\mathbf{S}_P\oplus_g\mathbf{S}_Q\rangle^g.$$

Let $\mathbf{A}_1 = \mathrm{Log}_{\mathbf{I}_{n,p}}^{gr}\left((\widetilde{\ominus}_{gr}\mathbf{U}_P\widetilde{\oplus}_{gr}\mathbf{U}_X)(\widetilde{\ominus}_{gr}\mathbf{U}_P\widetilde{\oplus}_{gr}\mathbf{U}_X)^T\right)$, $\mathbf{B}_1 = \mathrm{Log}_{\mathbf{I}_{n,p}}^{gr}\left((\widetilde{\ominus}_{gr}\mathbf{U}_P\widetilde{\oplus}_{gr}\mathbf{U}_Q)(\widetilde{\ominus}_{gr}\mathbf{U}_P\widetilde{\oplus}_{gr}\mathbf{U}_Q)^T\right)$, $\mathbf{A}_2 = \mathrm{Log}_{\mathbf{I}_n}^g(\ominus_g\mathbf{S}_P\oplus_g\mathbf{S}_X)$, and $\mathbf{B}_2 = \mathrm{Log}_{\mathbf{I}_n}^g(\ominus_g\mathbf{S}_P\oplus_g\mathbf{S}_Q)$. Then we have

$$\overline{\mathbf{Q}} = \operatorname*{arg\,max}_{\mathbf{Q}\in\mathcal{H}_{\mathbf{W},\mathbf{P}}^{psd,g}\setminus\{\mathbf{P}\}}\frac{\lambda\langle\mathbf{A}_1,\mathbf{B}_1\rangle_F + \langle\mathbf{A}_2,\mathbf{B}_2\rangle_F}{\sqrt{\lambda\|\mathbf{A}_1\|_F^2 + \|\mathbf{A}_2\|_F^2}\cdot\sqrt{\lambda\|\mathbf{B}_1\|_F^2 + \|\mathbf{B}_2\|_F^2}}$$
$$= \operatorname*{arg\,max}_{\mathbf{Q}\in\mathcal{H}_{\mathbf{W},\mathbf{P}}^{psd,g}\setminus\{\mathbf{P}\}}\frac{\langle[\sqrt{\lambda}\mathbf{A}_1\|\mathbf{A}_2], [\sqrt{\lambda}\mathbf{B}_1\|\mathbf{B}_2]\rangle_F}{\|[\sqrt{\lambda}\mathbf{A}_1\|\mathbf{A}_2]\|_F\cdot\|[\sqrt{\lambda}\mathbf{B}_1\|\mathbf{B}_2]\|_F}, \tag{9}$$

where $\|$ is the concatenation operation similar to operation $\mathrm{concat}_{spd}(.)$.

From the equation of hypergyroplanes in structure space $\widetilde{\mathrm{Gr}}_{n,p}\times\mathrm{Sym}_p^+$,

$$\langle\ominus_{psd,g}\mathbf{P}\oplus_{psd,g}\mathbf{Q}, \mathbf{W}\rangle^{psd,g} = 0.$$

Let $\mathbf{W} = (\mathbf{U}_W, \mathbf{S}_W)$. Then we have

$$\lambda\langle(\widetilde{\ominus}_{gr}\mathbf{U}_P\widetilde{\oplus}_{gr}\mathbf{U}_Q)(\widetilde{\ominus}_{gr}\mathbf{U}_P\widetilde{\oplus}_{gr}\mathbf{U}_Q)^T, \mathbf{U}_W(\mathbf{U}_W)^T\rangle^{gr} + \langle\ominus_g\mathbf{S}_P\oplus_g\mathbf{S}_Q, \mathbf{S}_W\rangle^g = 0. \tag{10}$$

Let $\mathbf{W}_1 = \mathrm{Log}_{\mathbf{I}_{n,p}}^{gr}\left(\mathbf{U}_W(\mathbf{U}_W)^T\right)$, $\mathbf{W}_2 = \mathrm{Log}_{\mathbf{I}_{n,p}}^g(\mathbf{S}_W)$. Then Eq. (10) can be rewritten as

$$\lambda\langle\mathbf{B}_1,\mathbf{W}_1\rangle_F + \langle\mathbf{B}_2,\mathbf{W}_2\rangle_F = 0,$$

which is equivalent to

$$\langle[\sqrt{\lambda}\mathbf{B}_1\|\mathbf{B}_2], [\sqrt{\lambda}\mathbf{W}_1\|\mathbf{W}_2]\rangle_F = 0. \tag{11}$$

Now, the problem in (9) is to find the minimum angle between the vector $[\sqrt{\lambda}\mathbf{A}_1\|\mathbf{A}_2]$ and the Euclidean hyperplane described by Eq. (11). The pseudo-gyrodistance from $\mathbf{X}$ to $\mathcal{H}_{\mathbf{W},\mathbf{P}}^{psd,g}$ thus can be obtained as

$$\bar{d}(\mathbf{X}, \mathcal{H}_{\mathbf{W},\mathbf{P}}^{psd,g}) = \frac{\langle[\sqrt{\lambda}\mathbf{A}_1\|\mathbf{A}_2], [\sqrt{\lambda}\mathbf{W}_1\|\mathbf{W}_2]\rangle_F}{\|[\sqrt{\lambda}\mathbf{W}_1\|\mathbf{W}_2]\|_F}$$
$$= \frac{\lambda\langle\mathbf{A}_1,\mathbf{W}_1\rangle_F + \langle\mathbf{A}_2,\mathbf{W}_2\rangle_F}{\sqrt{\lambda\|\mathbf{W}_1\|_F^2 + \|\mathbf{W}_2\|_F^2}}.$$

Some simple manipulations lead to

$$\bar{d}(\mathbf{X}, \mathcal{H}_{\mathbf{W},\mathbf{P}}^{psd,g}) = \frac{|\lambda\langle(\widetilde{\ominus}_{gr}\mathbf{U}_P\widetilde{\oplus}_{gr}\mathbf{U}_X)(\widetilde{\ominus}_{gr}\mathbf{U}_P\widetilde{\oplus}_{gr}\mathbf{U}_X)^T, \mathbf{U}_W\mathbf{U}_W^T\rangle^{gr} + \langle\ominus_g\mathbf{S}_P\oplus_g\mathbf{S}_X, \mathbf{S}_W\rangle^g|}{\sqrt{\lambda(\|\mathbf{U}_W\mathbf{U}_W^T\|^{gr})^2 + (\|\mathbf{S}_W\|^g)^2}},$$

which concludes the proof of Theorem 3.11.

$\square$

## N   PROOF OF PROPOSITION 3.12

*Proof.* We need the following result from Nguyen & Yang (2023).

**Proposition N.1.** *Let $M$ and $N$ be two Riemannian manifolds. Let $\phi : M \to N$ be an isometry. Then*

$$\mathrm{Log}_{\mathbf{P}}(\mathbf{Q}) = (D\phi_{\phi(\mathbf{P})}^{-1})(\widetilde{\mathrm{Log}}_{\phi(\mathbf{P})}(\phi(\mathbf{Q}))),$$

*where $\mathbf{P}, \mathbf{Q} \in M$, $D\tau_{\mathbf{R}}(\mathbf{W})$ denotes the directional derivative of a mapping $\tau$ at point $\mathbf{R} \in N$ along direction $\mathbf{W} \in T_{\mathbf{R}}N$, $\mathrm{Log}(.)$ and $\widetilde{\mathrm{Log}}(.)$ are the logarithmic maps in manifolds M and N, respectively.*

We adopt the notations in Bendokat et al. (2020). The Riemannian metric $g_{\mathbf{Q}}^O(.,.)$ on $O_n$ is the standard inner product given (Edelman et al., 1998; Bendokat et al., 2020) as

$$g_{\mathbf{Q}}^O(\Omega_1, \Omega_2) = \mathrm{Tr}(\Omega_1^T \Omega_2),$$

where $\mathbf{Q} \in O_n$, $\Omega_1, \Omega_2 \in T_{\mathbf{Q}} O_n$.

Let $\mathbf{U} \in \widetilde{\mathrm{Gr}}_{n,p}$, $\mathbf{D}_1, \mathbf{D}_2 \in T_{\mathbf{U}}\widetilde{\mathrm{Gr}}_{n,p}$. The canonical metric $g_{\mathbf{U}}^{\widetilde{\mathrm{Gr}}}(\mathbf{D}_1, \mathbf{D}_2)$ on $\widetilde{\mathrm{Gr}}_{n,p}$ is the restriction of the Riemannian metric $g_{\mathbf{Q}}^O(.,.)$ to the horizontal space of $T_{\mathbf{Q}} O_n$ (multiplied by 1/2) and is given (Edelman et al., 1998; Bendokat et al., 2020) by

$$g_{\mathbf{U}}^{\widetilde{\mathrm{Gr}}}(\mathbf{D}_1, \mathbf{D}_2) = \mathrm{Tr}\left(\mathbf{D}_1^T(\mathbf{I}_n - \frac{1}{2}\mathbf{U}\mathbf{U}^T)\mathbf{D}_2\right). \tag{12}$$

Let $\mathbf{P} \in \mathrm{Gr}_{n,p}$, $\Delta_1, \Delta_2 \in T_{\mathbf{P}} \mathrm{Gr}_{n,p}$. The canonical metric $g_{\mathbf{P}}^{Gr}(\Delta_1, \Delta_2)$ on $\mathrm{Gr}_{n,p}$ is the restriction of the Riemannian metric $g_{\mathbf{Q}}^O(.,.)$ to the horizontal space of $T_{\mathbf{Q}} O_n$ (multiplied by 1/2) and is given (Edelman et al., 1998; Bendokat et al., 2020) by

$$g_{\mathbf{P}}^{Gr}(\Delta_1, \Delta_2) = \frac{1}{2}\mathrm{Tr}\left((\Delta_{1,\mathbf{Q}}^{hor})^T \Delta_{2,\mathbf{Q}}^{hor}\right),$$

where $\Delta_{1,\mathbf{Q}}^{hor}$ and $\Delta_{2,\mathbf{Q}}^{hor}$ are the horizontal lifts of $\Delta_1$ and $\Delta_2$ to $\mathbf{Q}$, respectively. Here, $\mathbf{Q}$ is related to $\mathbf{P}$ by $\mathbf{Q} = (\mathbf{U}\ \mathbf{U}_\perp)$ and $\mathbf{P} = \mathbf{U}\mathbf{U}^T$, where $\mathbf{U} \in \widetilde{\mathrm{Gr}}_{n,p}$ and $\mathbf{U}_\perp$ is the orthogonal completion of $\mathbf{U}$.

Denote by $\mathrm{Hor}_{\mathbf{U}} \widetilde{\mathrm{Gr}}_{n,p}$ the horizontal space of $T_{\mathbf{U}}\widetilde{\mathrm{Gr}}_{n,p}$. Then this subspace is characterized by

$$\mathrm{Hor}_{\mathbf{U}} \widetilde{\mathrm{Gr}}_{n,p} = \{\mathbf{U}_\perp \mathbf{B} | \mathbf{B} \in \mathrm{M}_{n-p,p}\}.$$

From Eq. (3.2) in Bendokat et al. (2020),

$$g_{\mathbf{P}}^{Gr}(\Delta_1, \Delta_2) = \mathrm{Tr}\left((\Delta_{1,\mathbf{U}}^{hor})^T \Delta_{2,\mathbf{U}}^{hor}\right), \tag{13}$$

where $\Delta_{1,\mathbf{U}}^{hor}$ and $\Delta_{2,\mathbf{U}}^{hor}$ are the horizontal lifts of $\Delta_1$ and $\Delta_2$ to $\mathbf{U}$, respectively.

Therefore, by Eq. (12),

$$\begin{aligned}
g_{\mathbf{U}}^{\widetilde{\mathrm{Gr}}_{n,p}}(\Delta_{1,\mathbf{U}}^{hor}, \Delta_{2,\mathbf{U}}^{hor}) &= \mathrm{Tr}\left((\Delta_{1,\mathbf{U}}^{hor})^T(\mathbf{I}_n - \frac{1}{2}\mathbf{U}\mathbf{U}^T)\Delta_{2,\mathbf{U}}^{hor}\right) \\
&= \mathrm{Tr}\left((\Delta_{1,\mathbf{U}}^{hor})^T \Delta_{2,\mathbf{U}}^{hor}\right) - \frac{1}{2}\mathrm{Tr}\left((\Delta_{1,\mathbf{U}}^{hor})^T\mathbf{U}\mathbf{U}^T\Delta_{2,\mathbf{U}}^{hor}\right) \\
&= \mathrm{Tr}\left((\Delta_{1,\mathbf{U}}^{hor})^T \Delta_{2,\mathbf{U}}^{hor}\right) - \frac{1}{2}\mathrm{Tr}\left((\mathbf{U}_\perp\mathbf{B}_1)^T\mathbf{U}\mathbf{U}^T\mathbf{U}_\perp\mathbf{B}_2\right) \\
&= \mathrm{Tr}\left((\Delta_{1,\mathbf{U}}^{hor})^T \Delta_{2,\mathbf{U}}^{hor}\right) - \frac{1}{2}\mathrm{Tr}\left(\mathbf{B}_1^T\mathbf{U}_\perp^T\mathbf{U}\mathbf{U}^T\mathbf{U}_\perp\mathbf{B}_2\right) \\
&= \mathrm{Tr}\left((\Delta_{1,\mathbf{U}}^{hor})^T \Delta_{2,\mathbf{U}}^{hor}\right),
\end{aligned} \tag{14}$$

where the last equality follows from the fact that $\mathbf{U}^T\mathbf{U}_\perp = 0$.

Combining Eqs. (13) and (14), we get

$$g_{\mathbf{P}}^{Gr}(\Delta_1, \Delta_2) = g_{\mathbf{U}}^{\widetilde{\mathrm{Gr}}_{n,p}}(\Delta_{1,\mathbf{U}}^{hor}, \Delta_{2,\mathbf{U}}^{hor}).$$

By Proposition N.1,

$$\mathrm{Log}^{gr}_{\mathbf{P}}(\mathbf{F}) = (D\tau_{\tau^{-1}(\mathbf{P})})(\widetilde{\mathrm{Log}}^{gr}_{\tau^{-1}(\mathbf{P})}(\tau^{-1}(\mathbf{F}))),$$

where $\mathbf{P}, \mathbf{F} \in \mathrm{Gr}_{n,p}$.

From Eq. (3.15) in Bendokat et al. (2020),

$$D\tau_{\mathbf{R}}(\mathbf{W}) = \mathbf{R}\mathbf{W}^T + \mathbf{W}\mathbf{R}^T.$$

Therefore

$$\mathrm{Log}^{gr}_{\mathbf{P}}(\mathbf{F}) = \tau^{-1}(\mathbf{P})\big(\widetilde{\mathrm{Log}}^{gr}_{\tau^{-1}(\mathbf{P})}(\tau^{-1}(\mathbf{F}))\big)^T + \widetilde{\mathrm{Log}}^{gr}_{\tau^{-1}(\mathbf{P})}(\tau^{-1}(\mathbf{F}))\tau^{-1}(\mathbf{P})^T,$$

which concludes the proof of Proposition 3.12.

$\square$

