# OpenReview forum: "Matrix Manifold Neural Networks++"
_ICLR.cc/2024/Conference — ICLR 2024 poster_

### Official Review · Reviewer_iDyb · 2023-10-31

**Soundness:** 3 good
**Presentation:** 3 good
**Contribution:** 2 fair
**Rating:** 6
**Confidence:** 4

**Summary:**

- This paper discusses the design of deep neural networks (DNNs) on matrix manifolds, specifically Symmetric Positive Definite (SPD) and Grassmann manifolds.
- This paper mathematically proposes a way to compute the Fully Connected layers and Convolutional layers for SPD matrices.
- The paper also presents a method for performing back-propagation with the Grassmann logarithmic map in the projector perspective for Grassmann manifolds.
- Experiments are designed in human action recognition and node classification tasks, while both are very small datasets.
- The paper tries to address limitations of existing works and provides necessary mathematical tools for building DNNs on SPD and Grassmann manifolds.

----
I've read the rebuttal and would like to slightly increase the rating.
Thanks.

**Strengths:**

- The paper builds upon the theory of gyrovector spaces and extends it to matrix manifolds such as SPD and Grassmann manifolds.
- This paper defines a new way to build the basic blocks of neural networks - fully-connected and convolutional layers, for SPD matrices, and most specially for Grassmann, which is rarely discovered in the field.
- The authors demonstrate the mathematical rigor and effectiveness of their approach.
- The authors provide an ablation study and comparison against state-of-the-art methods, further validating the effectiveness of their approach.

**Weaknesses:**

- The experimental evaluation is based on some small and not generally used dataset, (not as big as ImageNet or equivalent). This limits the overall generalizability of the proposed approach.
- The network structures are limited with FC and CNN in most cases, while other network structures are missing, such as attention/ activation/ etc.
- It would be helpful if the author can clearly highlight the novel contributions and how they differ from or improve upon the existing theories discussed in other papers, for example [1,2,3,4]. Most of these existing papers discussed the building blocks of SPD network/ Grassmann network/ etc. Some other paper discussed other manifold such as Hyperbolic [5,6].



[1] Building Deep Networks on Grassmann Manifolds. Zhiwu Huang, Jiqing Wu, and Luc Van Gool.

[2] ManifoldNet: A Deep Neural Network for Manifold-valued Data with Applications. Rudrasis Chakraborty, Jose Bouza, Jonathan H. Manton, and Baba C. Vemuri.

[3] Dilated Convolutional Neural Networks for Sequential Manifold-valued Data. Xingjian Zhen, Rudrasis Chakraborty, Nicholas Vogt, Barbara B Bendlin, and Vikas Singh

[4] Neural Architecture Search of SPD Manifold Networks. Rhea Sanjay Sukthanker, Zhiwu Huang, Suryansh Kumar, Erik Goron Endsjo, Yan Wu, and Luc Van Gool.

[5] Hyperbolic Neural Networks. Octavian Ganea, Gary Becigneul, and Thomas Hofmann

[6] Hyperbolic Attention Network. Caglar Gulcehre, Misha Denil, Mateusz Malinowski, Ali Razavi, Razvan Pascanu, Karl Moritz Hermann, Peter Battaglia, Victor Bapst, David Raposo, Adam Santoro, and Nando de Freitas

**Questions:**

- What is the main difference between this paper and other existing manifold network?
- The author mentioned "our convolution operation can be used for dimensionality reduction", can you explain this more? From my understanding, if the input is NxN SPD matrix, the output has to lie within this NxN space. So I'm a little confusing about the reduction.
- Some tiny comments, the author defined exp(P) and log(P) but seems not use this annotation at all.

---

> ### Author Response · Authors · 2023-11-19
>
> We thank the reviewer for the feedback and suggestions for improving our work.
>
> All the datasets are popular in related literature (please refer to our general response and Tables 7, 8, and 9). We remarked that they are used in the recent work of Katsman et al. (2023) that is closely related to ours (see our response to reviewer 2o5X for the reference, we were not aware of it at the time of submission).
>
> Our work provides most ingredients for building attention blocks. A multi-head attention block includes:
> - FC layers: See Sections 3.2.1.
> - Concatenation operation: See Section 3.2.2.
> - Split operation can be performed as
> $\mathbf{P}_i = \operatorname{Exp}^g\_{\mathbf{I}\_{n\_i}}(\mathbf{A}\_i),$ $\mathbf{A}\_i \in \operatorname{M}\_{n\_i,n\_i},i=1,\ldots,N$ are given as
> \begin{equation*}
> \begin{bmatrix} \mathbf{A}\_{1} \cdots \cdots \\\ \cdots \mathbf{A}\_{2} \cdots \\\ \cdots \cdots \mathbf{A}\_{N}  \end{bmatrix} = \operatorname{Log}^g\_{\mathbf{I}\_n}(\mathbf{P}),
> \end{equation*}
> where $\mathbf{P}$ is input matrix, $n = \sum\_{i=1}^N n\_i$.
> - Similarity function (for computing attention weights): Distance-based function.
> - Activation function (for computing attention weights): Softmax.
> - Midpoint operation (for aggregating features): weighted Fréchet mean (wFM) in Log-Euclidean or Log-Cholesky framework.
>
> Multi-head attention can then be built, e.g. as in Shimizu et al. (2021).
>
> We proposed a non-linear activation in Grassmann networks in Section 3.4.2.
>
> Please refer to our general response for our contributions. The differences between our work and [1,2,3,4] are:
> - None of the works [1,2,3,4] uses the theory of gyrovector spaces to develop neural networks. One of the major advantages of a gyrovector space approach is that new concepts on matrix manifolds can be introduced to build neural networks. These concepts usually enjoy nice interpretations as their Euclidean and hyperbolic counterparts.
> - It is not trivial for [1,2,3,4] to extend MLR to matrix manifolds due to the lack of concepts like hyperplane.
> - In [1,2,3,4], some layers do not have an interpretation or their interpretation is not the same as ours. The interpretation of our FC layers (see the bottom of page 3) is fully analogous to those of FC layers in Euclidean DNNs and HNNs. FC layers are missing in [2,3]. FRMap and Bimap layers in [1,4] are called FC convolution-like layers (Huang & Gool, 2017; Huang et al., 2018). However, a Bimap layer takes an SPD matrix as input and produces a new SPD matrix as output (it can take a set of SPD matrices as input, but it produces a set of new SPD matrices, each of them from an input matrix). Thus, Bimap layers are not natural extensions of conv layers. The same issue applies to FRMap layers. FRMap and Bimap layers do not have an interpretation as our FC layers.
> - The conditions under which our method and theirs can be applied are different. The use of wFM in [2] results in some limitations, e.g. , input points of the convolution must lie in a ball of radius $r$, where $r$ is the convexity radius of the manifold. Our conv layers are built from the inverse and binary operations, inner product, and concatenation operation which are defined from the exponential and logarithmic maps.
>
> Thus, our building blocks are distinct from existing ones in many aspects, e.g., their interpretations and properties, conditions under which the method can be applied, computational complexity,...
>
> The differences between [5] and our work are:
> - Our MLR is based on the pseudo-gyrodistance, while MLR in [5] is based on the gyrodistance. Although the pseudo-gyrodistance does not have the same properties as the gyrodistance, our MLR still enjoys a nice interpretation (Nguyen
> & Yang, 2023, Fig. 2) that is fully analogous to that of the distance from a point to a Euclidean hyperplane.
> - Our method can build MLR on matrix manifolds that are not gyrovector spaces, while MLR in [5] is built on gyrovector spaces.
> - Conv layers are missing in [5].
>
> Results in Tables 7, 8, and 9 show that GyroSpd++ is superior to HypGRU in [5].
>
> [6] does not develop FC and conv layers as in our work. Like [5], techniques in [6] can only be applied to hyperbolic space.
>
> Note that extensions of building blocks in HNNs for matrix manifolds are usually not straightforward since matrix manifolds only have the gyro-structure (if this is the case) in a loose sense (Nguyen, 2022b). For instance, there exists
> a formula (Ganea et al., 2018) for the gyrodistance between a point and a hypergyroplane in hyperbolic space. However, such a formula does not exist (Nguyen & Yang, 2023) for SPD manifolds with Affine-Invariant metrics.
>
> The output of an FC layer can be computed as $\mathbf{Y} = \exp\big([y\_{(i,j)}]\_{i,j=1}^m\big)$,
> where $[y\_{(i,j)}]\_{i,j=1}^m$ is a matrix of size $m \times m$ (see Proposition 3.4). Our conv layers are built in a similar way as FC layers (see our response to reviewer pWPq). One can set $m < n$ to achieve dimensionality reduction.

---

### Official Review · Reviewer_pWPq · 2023-10-31

**Soundness:** 4 excellent
**Presentation:** 2 fair
**Contribution:** 4 excellent
**Rating:** 8
**Confidence:** 4

**Summary:**

This paper proposes novel formulations of fully-connected and convolutional layers, as well as MLR, on the SPD manifold based on gyrovector calculus.

**Strengths:**

The paper is well written, easy to follow, and the mathematics appear sound. The theoretical contributions are extensive and significant, with the potential of impacting research on matrix manifold neural networks in different areas.

Showing how to compute the Grassman logarithmic map in a differentiable way is another contribution.

The experimental evaluation for action recognition includes existing SPD deep learning methods and shows improvements.

**Weaknesses:**

The formulation of convolutional layers is more like a sketch of the proof than an actual definition.

The experimental evaluation for node classification as shown in the main text is fairly weak: the authors did not include any baseline, at least a few Euclidean-featured GNNs should have been included, as well as hyperbolic graph neural networks (Chami et al., and newer architectures).

There is actually no definition of what GyroSpd++ in the paper beyond a description of the matrix dimensions. Likewise, Gr-GCN is not properly defined in the experimental evaluation. Overall, this makes it difficult to understand the full network design and what is being evaluated.

**Questions:**

What non linearities were used? Is it a classical ReLU applied on the manifold, in the tangent space, or a specifically adapted rectifier such as those typically used with SPD deep nets, e.g., to inflate the small eigenvalues?

Can the authors restructure the text to improve the flow of the exposition and the introduction of the networks for the experimental evaluation?

As it stands, a lot of the content is in the supplementary.

---

> ### Author Response · Authors · 2023-11-19
>
> We thank the reviewer for the positive feedback and suggestions for improving the exposition of the paper.
>
> Concerning the definition of our convolutional layers, their output is computed in a similar way as the output of an FC layer. The only difference is that in the equation $v\_{(i,j)}(\mathbf{X}) = \langle \ominus_{ai}\mathbf{P}\_{(i,j)} \oplus\_{ai} \mathbf{X}, \mathbf{W}\_{(i,j)} \rangle^{ai}$ (see Proposition 3.4), $\mathbf{P}\_{(i,j)}$, $\mathbf{X}$, and $\mathbf{W}\_{(i,j)}$ are now obtained from operation $\operatorname{concat}\_{spd}(.)$ for three sets of SPD matrices. We propose to add a formal definition of our convolutional layers before the last paragraph of Section 3.2.2.
>
> Regarding our experimental evaluation for node classification, please refer to Table 14 in Appendix E.3.2 where we compare Gr-GCN++ against some baselines in SPD manifolds, Euclidean space, and hyperbolic space (including the method of Chami et al. (2019)).
>
> Regarding the architectures of GyroSpd++ and Gr-GCN, we could not give detailed descriptions for them in the paper due to space limit. GyroSpd++ has a MLR layer stacked on top of a convolutional layer (see the first paragraph of Section 4.1.1). For Gr-GCN, it is an extension of a standard GCN where feature transformation, aggregation, and bias and nonlinearity operations are performed on Grassmann manifolds (see Section 3.4.2). We propose to add graphic illustrations of these networks including that of GyroSpsd++ in the paper as follows:
> - A graphic illustration of Gr-GCN++ in Section 3.4.2.
> - Two graphic illustrations of GyroSpd++ and GyroSpsd++ in Section 4.1.1.
>
> We will restructure the text accordingly to improve the presentation of Section 4.
>
> Regarding the non-linear activations, we do not use them in our SPD neural networks since we did not observe any benefits of using them in our experiments (we tested with the ones in Huang & Gool (2017) and Nguyen (2022b)). For our network Gr-GCN++, our proposed activation function is given by $\sigma(\mathbf{X}) = \mathbf{V} \mathbf{V}^T$, where $\exp([\operatorname{Log}^{gr}\_{\mathbf{I}\_{n,p}}(\mathbf{X}), \mathbf{I}\_{n,p}]) \widetilde{\mathbf{I}}\_{n,p} = \mathbf{V} \mathbf{U}$ is a QR decomposition of $\exp([\operatorname{Log}^{gr}\_{\mathbf{I}\_{n,p}}(\mathbf{X}), \mathbf{I}_{n,p}]) \widetilde{\mathbf{I}}\_{n,p}$ (see Section 3.4.2).

---

### Official Review · Reviewer_2o5X · 2023-11-05

**Soundness:** 2 fair
**Presentation:** 2 fair
**Contribution:** 2 fair
**Rating:** 3
**Confidence:** 5

**Summary:**

The paper proposed a gyro space and gyro-vector based DNN on Grassmannian and SPD. The authors proposed formulation of building blocks like Convolution, Fully connected based on formulation derived in Nguyen, 2022a;b. The authors showed improved performance on human action recognition and node classification tasks formulated as on SPD and Grassmannian.

**Strengths:**

1. The paper is nicely motivated in the context of gyro vector representation.
2. The formulation of convolution and fully connected layers are nicely derived and formulated from Nguyen, 2022a;b.

**Weaknesses:**

1. The paper mostly based on formulation by Nguyen, 2022a;b. I don't want to treat this as a weakness, but a lack of strength.
2. The experiment results are rather naive, in recent years there is increasing literature in manifold DNNs so one wants to see thorough experimentations both in terms of different setting. Also some comparisons are missing including that with Chakraborty et al. (2020).

**Questions:**

The authors should elaborate what are the limitations in Chakraborty et al (2020) as they stated “Their proposed layers have nice theoretical prop- erties. A common limitation of the above works is that they do not provide necessary mathematical tools for constructing many essential building blocks of DNNs on SPD manifolds” without mentioning what mathematical building blocks are missing. A quick search reveals that same authors proposed other building blocks such as normalization using similar tools, please see “https://arxiv.org/pdf/2003.13869.pdf"
In 3.1, it is quite confusing why authors choose different notation to denote Grassmannian using different representations.
The authors should explain the abbreviation of the metrics on SPD like ai, le, lc used in section 3.1. For a general reader it will be easier to understand.
As FC layer can be treated as special case of convolution (with full kernel size), the authors should restructure the paper by defining convolution first, then the authors can just address FC trivially.
It is not clear why authors suddenly defining MLR in section 3.3 on positive semi-definite matrices. The MLR for SPD (and trivially can be extended to SPSD ) by using formulation in  Nguyen (2022a)
What is the motivation behind using different metrics: “We use Affine-Invariant metrics for the convolutional layer and Log-Euclidean metrics for the MLR layer”?
What is rationale behind better performance using structure space representation as mentioned in “These results show that MLR is effective when being designed in structure spaces from a gyrovector space perspective.”? They should essentially represent same space, why the difference in achieving different optima?

---

> ### Author Response · Authors · 2023-11-19
>
> We thank the reviewer for the feedback and suggestions for improving our work. Please refer to our general response and Tables 6, 7, 8, 9, and 14 in Appendix.
>
> We did not compare our work against Chakraborty et al. (2020) because our work is closer to Huang & Gool (2017); Brooks et al. (2019) than Chakraborty et al. (2020) that focusses on data that are sample points on a manifold. Chakraborty et al. (2020) stated that “In Huang & Gool (2017), authors are not dealing with manifold-valued images as input data...” This probably explains why Chakraborty et al. (2020) did not compare their network against SPDNet and SPDNetBN. We remark that the recent work of Katsman et al. (2023) that is closely related to ours only compares their SPD network against SPDNet and SPDNetBN. As requested, we are working on some experiments and will report results when they are done.
>
> Some missing blocks in Chakraborty et al. (2020) are FC layers, attention mechanisms, and MLR:
> - Their linear classifier cannot be used as an FC layer in a general context of their networks (and ours).
> - An FC layer can be seen as a linear map y = Ax + b, where A is the weight matrix, x, y, and b are the input, output, and bias vectors. Thus, in our framework, a natural extension of FC layers should take a point on the manifold as input and output a point on the manifold. If one treats FC layers as special case of conv layers with full kernel size, then “FC layers” can be built from their conv layers, but their input is now a set of points on the manifold.
> - Attention modules usually include FC layers which are missing in their work.
> - It is not trivial for their method to build MLR due to the lack of concepts like hyperplane.
>
> Batch norm layers can be built with a gyrovector approach using the binary and inverse operations, and scalar multiplication in Nguyen (2022b). Attention modules can also be developed (see our response to reviewer iDyb).
>
> We use different notations for Grassmann manifolds because we want to differentiate the domain and codomain of mapping $\tau$ in Proposition 3.12, and to clarify operations in Definitions 3.7 and 3.8.
>
> We already mentioned the abbreviations ai, le, lc  in Section 2.1.
>
> We construct FC and conv layers  using a similar approach as Shimizu et al. (2021). One can construct them as suggested. However, the resulting FC layers might no longer have the same interpretation as our layers. They might also not follow our above design.
>
> Our choice of metrics is done empirically (see Table 6, Appendix D.4.1). We select the best architecture on FPHA and use it for all experiments. This limitation has been stated in Appendix F. Note that Ganea et al. (2018) mentioned a similar issue.
>
> Our SPSD MLR is motivated by Nguyen & Yang (2023) and aims to address their work limitations:
> - Limited range of applicability: First, their formulation of hypergyroplanes relies on
> the logarithmic map which might not be well-defined in some cases. Second, their derivation of pseudo-gyrodistance requires some properties of SPD manifolds that are not satisfied by other manifolds, e.g., Grassmann or SPSD manifolds. Our Theorem 3.11 generalizes Theorems 2.23, 2.24, and 2.25 of Nguyen & Yang (2023)). For instance,
> when λ = 0 (full-rank matrices), one obtains a unified formula for the pseudo-gyrodistance instead of different formulae in Nguyen & Yang (2023). One can seamlessly translate our result to the case of Grassmann manifolds.
> - Computational cost: Our work is motivated by Bonnabel et al. (2013) (see Appendix D.3).
>
> In Nguyen (2022a), each SPSD matrix $\mathbf{P} = (\mathbf{U}\_P,\mathbf{S}\_P)$ is represented by $\mathbf{S}\_P$, while our formulation preserves both $\mathbf{U}\_P$ and $\mathbf{S}\_P$. Our experiments show that GyroSpsd++ based on Nguyen (2022a) gives 75.24 $\pm$ 1.28 and 92.39 $\pm$ 0.22 on HDM05 and FPHA, respectivly, while GyroSpsd++ gives 78.52 $\pm$ 1.34 and 97.90 $\pm$ 0.24, respectivly. This agrees with results in Nguyen (2022a) where full-rank models outperform low-rank ones.
>
> Both SPSD MLR and SPD MLR suffer from the numerical instability issue which is common to all layers based on SPD operations (SVD, eigen decomposition). One reason is because network inputs are usually ill-conditioned. We deal with this by using custom backprop (Ionescu et al., 2015) that only gives approximated gradients for SPD operations. This somehow affects performance. In our work, an SPSD matrix is represented by a subspace and an SPD matrix. Although the SPD matrix is not always positive definite in practice, we are able to remove the need for using custom backprop at certain steps, thus potentially improving performance. This probably explains the fact that GyroSpsd++ outperforms GyroSpd++ in some cases.
>
> **References**
>
> Isay Katsman, Eric Ming Chen, Sidhanth Holalkere, Anna Asch, Aaron Lou, Ser-Nam Lim, and Christopher De Sa. Riemannian Residual Neural Networks. CoRR, abs/2310.10013, 2023. URL https://arxiv.org/abs/2310.10013.

---

> > ### Author Response · Authors · 2023-11-21
> >
> > As requested, we would like to provide our experimental results for comparison of our method against the method of Chakraborty et al. (2020). We only use HDM05 and FPHA datasets due to time constraints.
> >
> > Our experiments are performed as follows. We replace the convolutional layer in GyroSpd++ with a convolutional layer (Conv-wFM layer) proposed in their work. The resulting network has a MLR layer (remind that the MLR layer in GyroSpd++ is based on a Log-Euclidean metric) stacked on top of a Conv-wFM layer. We also test another version of this network by stacking a MLR layer based on an Affine-Invariant metric on top of a Conv-wFM layer. We do not consider Log-Cholesky metrics due to their inferior performance compared to Affine-Invariant and Log-Euclidean metrics reported in Nguyen (2022b) (see also Table 6 in our paper). Given $\mathbf{P}\_i \in \operatorname{Sym}\_n^+,i=1,\ldots,N$, $w\_i \in \mathbb{R}, w_i > 0,i=1,\ldots,N$, the $i^{th}$ estimate $\mathbf{M}\_i$ of the weighted Fréchet mean of $\mathbf{P}\_i,i=1,\ldots,N$ is given (Chakraborty et al., 2020) by the following recursive formula:
> > \begin{equation*}
> > \mathbf{M}\_i = \delta\_{\mathbf{M}\_{i-1} \rightarrow \mathbf{P}\_i}\bigg(\frac{w\_i}{\sum\_{j=1}^i w\_j}\bigg),
> > \end{equation*}
> > where $\mathbf{M}\_1 = \mathbf{P}\_1$ and $\delta\_{\mathbf{X} \rightarrow \mathbf{Y}}: [0,1] \rightarrow \operatorname{Sym}\_n^+$ is the shortest geodesic curve from $\mathbf{X}$ to $\mathbf{Y}$. The condition that the weights are normalized is not required here since this will not change the result in the recursive formula.
> >
> > For each network, we implement two methods for computing the weighted Fréchet mean, i.e., the one above and another one based on Log-Euclidean metrics that admits a closed-form expression. Our implementation of the method in Chakraborty et al. (2020) follows closely the official code published by the authors at https://github.com/cvgmi/manifold-net-dmri. All the networks use the same input data as GyroSpd++ which is a set of SPD matrices given in Eq. (4) of Appendix D.2.2. The output of the Conv-wFM layer is thus the weighted Fréchet mean of these matrices. The networks are learned using the same settings as our network (see Appendix D.2.1). Results from Table A show that GyroSpd++ significantly outperforms its competitors on both the datasets. We also tested architectures with 2 Conv-wFM layers as it has been shown (Chakraborty et al., 2020) that Conv-wFM layers are most likely non-collapsible on non constant sectional curvature manifolds (e.g., SPD manifolds) and thus stacking multiple Conv-wFM layers can be beneficial. However, we did not observe any improvement with these architectures in our experiments.
> >
> > | Dataset | HDM05 | FPHA |
> > |-----------------------|-------------|-------------|
> > | Conv-wFM-MLR-LE (wFM based on Chakraborty et al. (2020) ) |    53.26 $\pm$ 1.46    |   80.16 $\pm$ 0.32    |
> > | Conv-wFM-MLR-AI  (wFM based on Chakraborty et al. (2020) ) |    54.21 $\pm$ 1.34    |   88.70 $\pm$ 0.29    |
> > | Conv-wFM-MLR-LE  (wFM based Log-Euclidean metrics )         |    53.47 $\pm$ 1.31    |   83.30 $\pm$ 0.26    |
> > | Conv-wFM-MLR-AI  (wFM based Log-Euclidean metrics )          |    55.68 $\pm$ 1.35    |   87.04 $\pm$ 0.28    |
> > | GyroSpd++ (Ours)                                                                        |    **79.78** $\pm$ 1.42    |   **96.84** $\pm$ 0.27    |
> >
> > Table A: Comparison of our method against the method of Chakraborty et al. (2020).

---

### Author Response · Authors · 2023-11-19
**General Response**

We thank the reviewers for their insightful comments and suggestions for improving our work.

Our contributions can be summarized as follows:
- We derive FC and convolutional layers for SPD neural networks. While some extensions of convolutional layers in the same setting have been developed in Huang & Gool (2017); Chakraborty et al. (2020), our approach is fundamentally different from them. More precisely, our method is based on the concept of SPD hypergyroplane and that of SPD pseudo-gyrodistance from an SPD matrix to an SPD hypergyroplane, while none of these concepts is used in the above works. As a result, our FC layers have an interpretation which is fully
analogous to those of FC layers in Euclidean and hyperbolic neural networks (Shimizu et al., 2021). Please refer to our response to reviewer iDyb for more details on the differences of our work and existing works.
- To the best of our knowledge, our work is the first that shows: (1) how to perform backpropagation with the Grassmann logarithmic map in the projector perspective without resorting to any approximation schemes, and (2) how to construct graph convolutional networks (GCNs) on Grassmann manifolds for node classification. These contributions do not seem to get enough consideration by some reviewers.
- We develop MLR on SPSD manifolds. MLR in Nguyen & Yang (2023) relies on the gyro-structure of SPD manifolds. This is one of the main limitations of their work since not many matrix manifolds have such a nice property. Our work addresses this limitation by developing MLR on matrix manifolds that are not gyrovector spaces. Our theoretical result in Theorem 3.11 generalizes several theoretical results in Nguyen & Yang (2023). Please refer to our response to reviewer 2o5X for more details.

Regarding our experiments, they are performed on six datasets with different characteristics:
- Human action recognition: We use three datasets containing three different action categories, i.e., human body actions (HDM05), hand actions (FPHA), and interaction actions (NTU60). HDM05 dataset is commonly used for comparing discriminative SPD neural networks (Huang & Gool, 2017; Brooks et al., 2019; Sukthanker et al., 2021; Nguyen, 2022a;b; Nguyen & Yang, 2023). FPHA dataset is widely used for comparing hand gesture recognition methods (Liu et al., 2020; Wang et al., 2023; Wen et al., 2023). NTU60 (10347 samples) is part of the large-scale NTU RGB+D 60 dataset that has been extensively used for comparing action recognition approaches.
- Node classification: We use three datasets, i.e., Airport, Pubmed, and Cora. These are among the most widely used datasets for the task (Kipf & Welling, 2017; Velic̆ković et al., 2018; Liu et al., 2019; Chami et al., 2019; Zhang et al., 2021; 2022; Kang et al., 2023).

We also made an effort to show the effectiveness of the proposed methods by comparing them against closely related methods (neural networks on manifolds) as well as state-of-the-art methods for the considered applications:
- Related methods: For human action recognition, we use SPDNet and SPDNetBN that are the reference SPD models as well as the most recent works based on a gyrovector space approach, i.e., GyroAI-HAUNet, SPSD-AI, and MLR-AI (see Table 1 for references). We also use a hyperbolic neural network (HypGRU). However, since HypGRU is inferior to the competing methods, we do not investigate more recent hyperbolic models for this application. For node classification, we use Euclidean graph neural networks (GCN and GAT), a graph neural network on SPD manifolds (SPD-GCN), graph neural networks on hyperbolic space (HGNN, HGCN, LGCN, and HGAT). Please refer to Table 14 in Appendix E.3.2 for references.
- State-of-the-art methods: For human action recognition, we use graph neural networks (MS-G3D and TGN), and a Transformer (ST-TR). Note that MS-G3D, TGN, and ST-TR are specifically designed for skeleton-based action recognition. Please refer to Tables 7, 8, and 9 in Appendix D.4.2 for references. For node classification, we use the recent network HamGNN (see Table 14).

In terms of experimental results:
- Human action recognition: Our networks outperform state-of-the-art methods on all the datasets where significant improvements can be observed in many cases. Please refer to Tables 7, 8, and 9 in Appendix D.4.2 for details.
- Node classification: Our network achieves the best and second best results on Pubmed and Cora dataset, respectively. Please refer to Table 14 in Appendix E.3.2 for details.

Please refer to Appendices for more experimental results.

**References**

Rui Wang, Xiao-Jun Wu, Tianyang Xu, Cong Hu, and Josef Kittler. U-SPDNet: An SPD Manifold
Learning-based Neural Network for Visual Classification. Neural Networks, 161:382–396, 2023.

Yilin Wen, Hao Pan, Lei Yang, Jia Pan, Taku Komura, and Wenping Wang. Hierarchical Temporal
Transformer for 3D Hand Pose Estimation and Action Recognition from Egocentric RGB Videos.
In CVPR, pp. 21243–21253, 2023.

---

### Meta-Review · Area_Chair_yZbj · 2023-12-02

**Metareview:**

This paper designs fully-connected and convolutional layers for SPD neural networks. How to perform backpropagation on SPD and Grassmann manifolds is also discussed. Superior results on human action recognition and node classification were obtained. These results are important to study neural networks over Riemannian manifolds.

**Justification For Why Not Higher Score:**

Though the results are important, they are not significant enough to receive a higher score.

**Justification For Why Not Lower Score:**

The results are important, the paper should be accepted.

---

### Decision · Program_Chairs · 2024-01-16

Accept (poster)